# Magma flow localisation during dyke propagation produces complex magma transport pathways

C. Allgood[1] ✉, E. W. Llewellin[2] ✉, R. J. Brown[2] & A. Loisel[3]

Basaltic fissure eruptions, the most common type of eruption on Earth, are fed by dykes, which are magma-filled cracks that propagate through the crust. It is well-established that dykes have a segmented structure, but the impact of this structural complexity on the development of magma flow pathways and on the behaviour of any subsequent eruptions remains largely unexplored. Here, we present field evidence from a solidified dyke in Tenerife (Canary Islands, Spain) that is exceptionally well-exposed, displaying segmentation structures that reveal complex, three-dimensional magma transport pathways. The dyke consists of plate-like lobes, and its layered internal textures record flow localisation, analogous to lava tube development in pahoehoe flow fields. We propose that flow localisation mediates magma supply to the leading edge of the propagating dyke, creating a convoluted plumbing system that likely influences eruption behaviour, and which should be accounted for in models of magma transport.

The structure of a propagating dyke influences its potential to reach the surface without stalling, and determines the location, behaviour and longevity of any subsequent fissure eruptions it may feed. For example, the likelihood of a flow pathway sealing shut is determined by its width[1], whereas the transport of volatiles will be influenced by a branching conduit structure[2]. Field studies of exposed, solidified dykes have shown that they are emplaced as a series of segments[3,4], and the fissure eruptions fed by dykes are similarly segmented[5], reflecting the complex structure supplying them. Information on eruption likelihood, location and behaviour is vital for hazard mitigation; however, there has been limited consideration of how the segmented structure of dykes influences the development of subsurface flow pathways, and how this may impact any subsequent eruption dynamics.

The segmented nature of dykes has been shown by field studies[4,6,7], and inferred from the volcano-tectonic seismicity that accompanies dyke emplacement[8–10]. Sills, which are bedding-parallel analogues to dykes, also show complex segmentation, as revealed by field studies[11–13] and seismic reflection surveys[14–16], and magma flow localisation within sill segments is evidenced by the existence of tube-like conduits[14–16]. Laboratory analogue experiments have also produced segmented intrusion structures in granular[17] or viscoelastic[18] host materials, and by using solidifying fluids[19–21], and flow localisation within segments has been reproduced with both isothermal[22,23] and solidifying[19–21] fluids. However, relating the findings of analogue experiments to field evidence is challenging, because solidified dykes are typically only exposed on one plane[24,25], which limits the interpretation of their three-dimensional structure. Field evidence for the three-dimensional organisation and flow dynamics of dyke segments is therefore crucial for verifying and inspiring analogue models, and for aiding the interpretation of seismic datasets.

In order to forecast eruption likelihood and behaviour based on active dyke structures inferred from seismicity[8–10], models must be developed that account for segmentation dynamics and their impact on magma flow. For example, the transport of volatiles would be impacted by a complex, segmented plumbing system: firstly, the large-scale convection expected within a single, vertically extensive planar volume is likely to be suppressed if this volume is divided into several smaller, interconnected segments; secondly, convective localisation may be enhanced by splitting a wide planar structure into several smaller ones[26]; and thirdly, branching conduits will cause volatile

[1]Lancaster Environment Centre, Lancaster University, Lancaster, UK. [2]Department of Earth Sciences, Durham University, Durham, UK. [3]Université de Strasbourg, CNRS, Institut Terre et Environnement de Strasbourg, Strasbourg, France. ✉e-mail: c.allgood1@lancaster.ac.uk; ed.llewellin@durham.ac.uk

segregation[2,27]. Such complex circulation patterns could potentially lead to different behaviour at individual vents in the same eruption, as seen at Tajogaite in 2021[28]. Models incorporating these processes must be informed by field evidence for three-dimensional dyke structure, and by rock textures evidencing magma flow conditions; for example, textures at dyke margins have revealed magma flow within segments to be complex, varying along exposed segment lengths[4,29–31].

Here, we present field evidence from a solidified dyke in Tenerife (Canary Islands, Spain), which is exceptionally well-exposed, displaying segments on both horizontal and vertical exposures. Opportunities to interpret the true, three-dimensional architecture of dykes are rare[32], so the dyke presented here provides great insight into dyke emplacement processes. Furthermore, this dyke contains nested, layered textures indicative of flow localisation, captured as the segment progressively solidified. By combining new field observations of segment relationships and internal textural layers with published evidence of dyke tip processes, segmentation and flow localisation, and by drawing parallels with localisation and tube formation within lava flows, we present a new conceptual model for dyke emplacement. This conceptual model will inform future laboratory and computational modelling efforts, and aid the interpretation of seismicity during active emplacement. We use the term "dyke architecture" to imply dynamic plumbing system construction, with dyke emplacement self-organising in response to its surroundings and internal flow processes.

## Results

### The Carrizales Dyke

We present a dyke from the Teno Massif, in northwest Tenerife, Spain (location in Supplementary Fig. 1). It is exposed on the ridge south of Los Carrizales, so we refer to it as the Carrizales Dyke. The Teno Massif is the deeply eroded remains of a basaltic shield volcano constructed between 6.3 and 5.1 Ma[33,34]. The precise height and position of the old edifice summit are unknown, as it suffered two major collapse events, evidenced by unconformities defining horseshoe-shaped troughs opening to the northeast of the region[34].

The Carrizales Dyke can be traced for 1500 m along the ridge. It is composite, containing three layers of distinct texture and composition. We focus on a well-exposed section comprising five segments (Fig. 1), the shortest of which has a lateral extent less than 10 m (S3), whereas the longest extends around 200 m (S2). Each segment contains the three compositional layers and has a near uniform width along its length (98 ± 5 cm average), only tapering within 2 m of its tips (Fig. 1d). The segments overlap in four relay zones, where the tips run parallel for between 1 m and 2.5 m, either directly in contact (Fig. 1d), or separated by a screen of host rock (Fig. 1e) or by an older dyke (Fig. 1f). The relay zones are therefore small in comparison to the apparent length of segments. We infer that the Carrizales Dyke was emplaced at relatively shallow depths, up to several hundred metres, as it contains vesicles from volatile exsolution. The dyke margins are undulous at a scale of centimetres, but there are no major variations in dyke width of the sort associated with intrusion into unconsolidated scoria[35]. We therefore infer that the dyke intruded into partly consolidated material, via a combination of brittle fracturing and viscous indentation[36].

The host rock on the ridge and in the surrounding valleys is dominantly volcaniclastic, composed of centimetre to decimetre sized clasts of basaltic lava and pyroclastic material, arranged in beds centimetres to decimetres thick. These beds dip steeply to the west, and are continuous over tens of metres, but pinch and swell in thickness. The host rock contains localised fractures, which cannot be traced more than 10 m, and which show minimal vertical offset. The age of these fractures relative to the Carrizales Dyke is unknown; other dykes on the ridge have fractures cutting across them, but the Carrizales Dyke does not. There is no evidence of the Carrizales Dyke interacting with fractures.

### Segment geometry and inferred propagation directions

The Carrizales Dyke displays segmentation on both horizontal and vertical exposures (Fig. 1). In field studies of dykes, it is commonly assumed that segments connect back to a continuous parent body[3,13]. If segments are disconnected on a horizontal plane, magma did not flow laterally between them at the point of observation, and so flow must have had a vertical component. Similarly, segments seen on vertical cliff faces are assumed to have formed due to a horizontal flow component. Therefore, the segmentation of the Carrizales Dyke, occurring on both horizontal and vertical exposures, demonstrates that the magma propagated with components of both vertical and lateral flow.

The evidence for propagation direction is clearest in S3, which is very limited in its horizontal and vertical extents (10 m and 25 m respectively). Evidently, the propagation of S3 was halted in the SW and downwards directions, which suggests that we are now observing a 'corner' of the segment (Fig. 2a). We can also infer that the NE edge of S4 is either vertical, or dips to the SW, because it doesn't crop out on the cliff containing S3.

The relay between S1 and S2 is of interest, as these segments terminate on either side of another dyke, D2, 0.6 m wide, which is vertically oriented (Fig. 1f). D2 is likely older, as its strike is unaltered by the presence of the Carrizales Dyke, whereas younger dykes are often deflected by older dykes[37]. It therefore appears that the Carrizales Dyke developed segments in response to D2. We do not know whether S1 and S2 are connected below, or were connected above, the point of observation, but both are likely to have vertical edges against D2 in this vicinity (Fig. 2b).

The S3-S4 relay is also of interest, as the two segments are directly in contact (Fig. 1d). If the segments had both been entirely molten, we would expect them to coalesce, as in the model of Pollard et al.[3]. The fact that these segments remain separate demonstrates that the margins had solidified, providing a barrier that resisted coalescence. It is possible that the initial magma pulse for one segment arrived before the other, allowing the earlier segment to solidify, and making coalescence with the later segment less likely.

The segment tips, where visible, generally have a tapering form with a few minor, centimetre-scale offshoots into the host rock (Fig. 1). The geometry of solidified segments cannot be taken to represent their active, molten state, as the segments are likely to have inflated and/or deflated while solidifying progressively inwards from their edges[7,38]. Likewise, the tapering nature of the segment tips does not represent the shapes of the active, propagating tips; instead, the solid tips capture a stage of emplacement after the segment inflated[31,36], which would suggest that the active tips were considerably narrower and more tapered when their propagation stalled.

### Textural layers and intrusion events

We now examine the rock textures within the dyke segments, to build on our inferences from the large-scale dyke structure. Dykes solidify progressively inwards from their margins, so their textures from margins to centre can be read as a time series[39–41]. The Carrizales Dyke is composite, and its texturally distinct layers are likely to be the result of three separate intrusion events: the outer layer displays marginal bands, similar to those described by Allgood et al[41].; the intermediate layer has a similar composition to the outer layer but is significantly more vesicular; and the central layer has no vesicles and contains clinopyroxene phenocrysts 2–8 mm in diameter (Fig. 3). The three layers are present in each segment within the study area. Measurements from the accessible regions of the dyke, mostly along S4, show that the marginal layers have an average thickness of 15 ± 6 cm, the intermediate layers are 14 ± 5 cm; and the central layer is typically 43 ± 5 cm. Layer thicknesses are the same for each segment. Thickness data from S4 are presented in Fig. 4 and are available as Supplementary data.

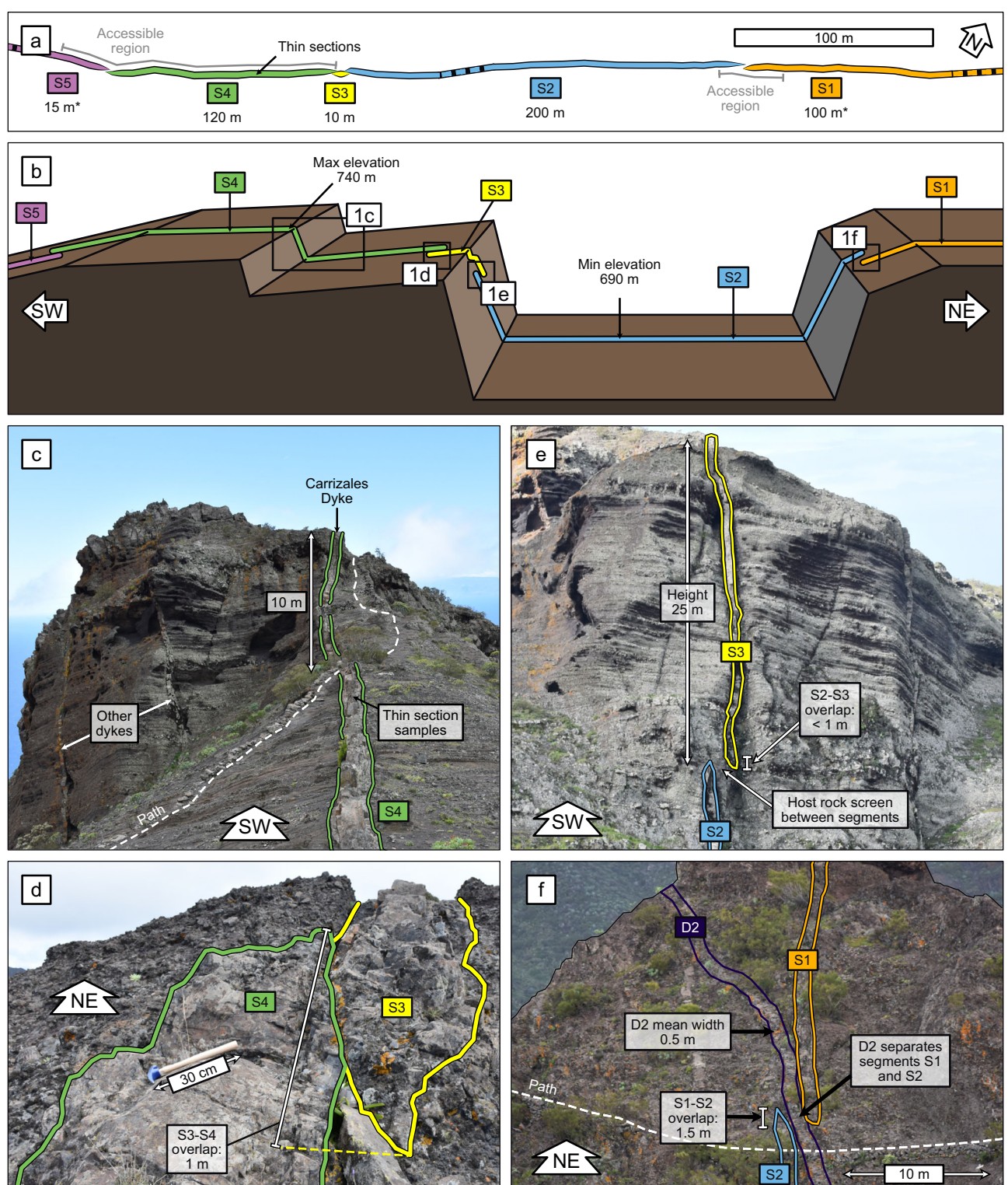

**Fig. 1 | Layout of the studied area of the Carrizales Dyke. a** Map view of segments, including segment lengths. An asterisk indicates that the segment is only partially exposed, so these lengths are less than the full lengths. **b** Schematic representation of segment exposures on the ridge. **c** View along S4, from S3-S4 relay. **d** S3-S4 relay on gentle SW-facing slope. **e** S2-S3 relay on vertical NE-facing cliff. **f** S1-S2 relay on steep SW-facing slope, with segments separated by another dyke, D2.

The marginal layer represents the initial intrusion in which the segments were established. The margins are undulous at a scale of several centimetres, curving around clasts in the host rock (Fig. 3a, b). At the margin, the banded textures follow the undulations closely, but they become straighter towards the dyke centre, gradually aligning with the dyke strike. We therefore infer that the initial crack was irregular, relative to its width, and that the dyke became more linear as it inflated.

There is a chilled margin between the intermediate and central layers, with a noticeable drop in grain size (Supplementary Fig. 10). This shows that the intermediate layer was cold enough to induce a chill on the central layer, suggesting that a significant amount of time

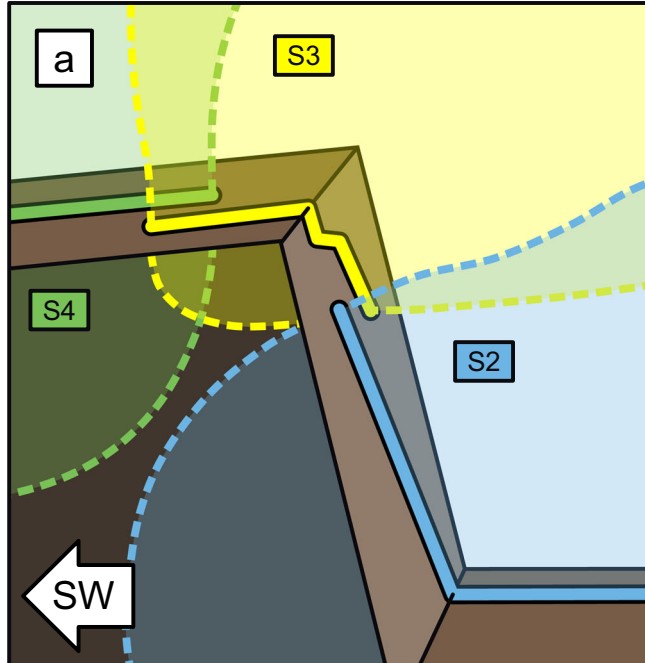

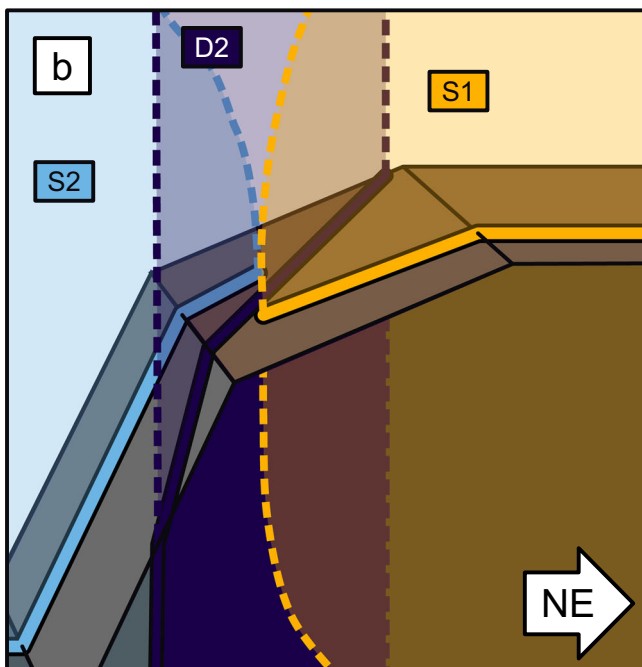

**Fig. 2 | Schematic representation of relations between segments. a** S3 exposures reveal limited extent of segment, most likely a lower 'corner', whereas NE edge of S4 is either vertical or dips SW. **b** Older vertical dyke D2 separates S1 and S2, which have vertical edges in this vicinity, but presumably connect somewhere outside the plane of observation.

elapsed between their emplacements[42]. However, the three intrusion events presumably occurred in quick enough succession that it remained energetically favourable for each intrusion to exploit the warm, weak interior of the previous layer[43].

## Sub-layering and flow localisation

In addition to the three main layers, the central layer of S4 contains sub-layers defined by the concentration of clinopyroxene phenocrysts. These are not true layers, as they lack chilled margins and show no change in groundmass texture (Fig. 3). The sub-layers are present in a 20-m-long section towards the SW end of S4, and they appear to be nested within one another (Fig. 4). As such, we infer that each sub-layer represents a batch of magma within the third intrusion event, with the same melt composition, but carrying a different phenocryst load from upstream.

The nested sub-layers within S4 are evidence of flow localisation during the propagation phase. Dykes solidify progressively inwards from their margins, and textural layers record distinct batches of magma transiting the system, with the youngest batch occupying the dyke centre. The nested sub-layers within S4 become progressively restricted in their lateral extent, demonstrating that each new batch of magma occupied a smaller region, thereby evidencing that flow localised over time (Fig. 4).

Flow localisation may be influenced by several processes. A leading mechanism for flow localisation is thermal feedback, which has been modelled extensively for dykes that feed fissure eruptions[1,44–49]. Similar feedback processes should be expected to occur throughout the dyke propagation phase. Thermal feedback operates as follows: magma has a temperature-dependent viscosity, so cooler sections of a dyke flow more slowly, while hotter sections advance more rapidly. This results in narrow, slow-flowing sections sealing shut while wider, fast-flowing sections remain open, meaning that the longevity of a channel is determined by dyke width and flow rate[1]. Localisation driven by thermal feedback may be enhanced by other localisation mechanisms, such as the fluid-dynamically mediated flow focussing observed in analogue experiments, in which Newtonian fluids injected upwards into gelatine have formed a fast-flowing jet in the centre of the fluid-filled fracture[22,23].

## Crystal imbrication and flow directions

Finally, we use small-scale rock textures to infer magma flow directions. Crystal imbrication is a texture produced by crystals stacking like fallen dominoes, pointing in the direction of flow[39]. By outlining phenocrysts in photographs and microlites in thin section, we capture the preferred orientation of crystals, which reflects the mechanical interactions that caused them to stack. We focus on preferred orientations within S4, which is accessible along its entire length. We use four sites where surfaces are fresh enough to reveal at least 30 phenocrysts in each half of the dyke. The exposures are approximately horizontal, so the preferred orientations indicate the lateral flow component, which is consistently to the SW throughout S4 (Fig. 4).

To determine the direction of the vertical flow component, we took samples from each layer of the dyke midway along S4, on the southern side (Fig. 1), and took thin sections in the vertical and horizontal planes, perpendicular to the dyke strike. The preferred orientation of plagioclase microlites in each layer indicates a SW lateral flow component and a weaker downwards vertical component (Fig. 4c). For the intermediate layer, the flow direction is also supported by the orientations of elongated vesicles (Supplementary Fig. 8d). Magma appears to have flowed dominantly SW with a downwards component for each intrusion event; however, the imbrication analysis provides no way to ascertain the steepness of the downwards flow component.

We infer that the crystal imbrication textures in S4 reflect the initial magma flow direction, rather than any subsequent downwelling at the end of the active intrusion phase[50,51]. Late-stage downwelling would be expected to overwrite any pre-existing lateral flow textures, but the observed imbrication is dominantly lateral. Furthermore, downwards imbrication is found at the dyke margin, where textures were captured within tens of minutes of the magma arriving[41], proving that the initial magma flow had a downwards component at this position. However, the dominant emplacement direction appears to have been lateral.

## Inferred architecture and emplacement

We now use our observations of segment structure, internal textures and inferred flow directions to visualise the architecture and

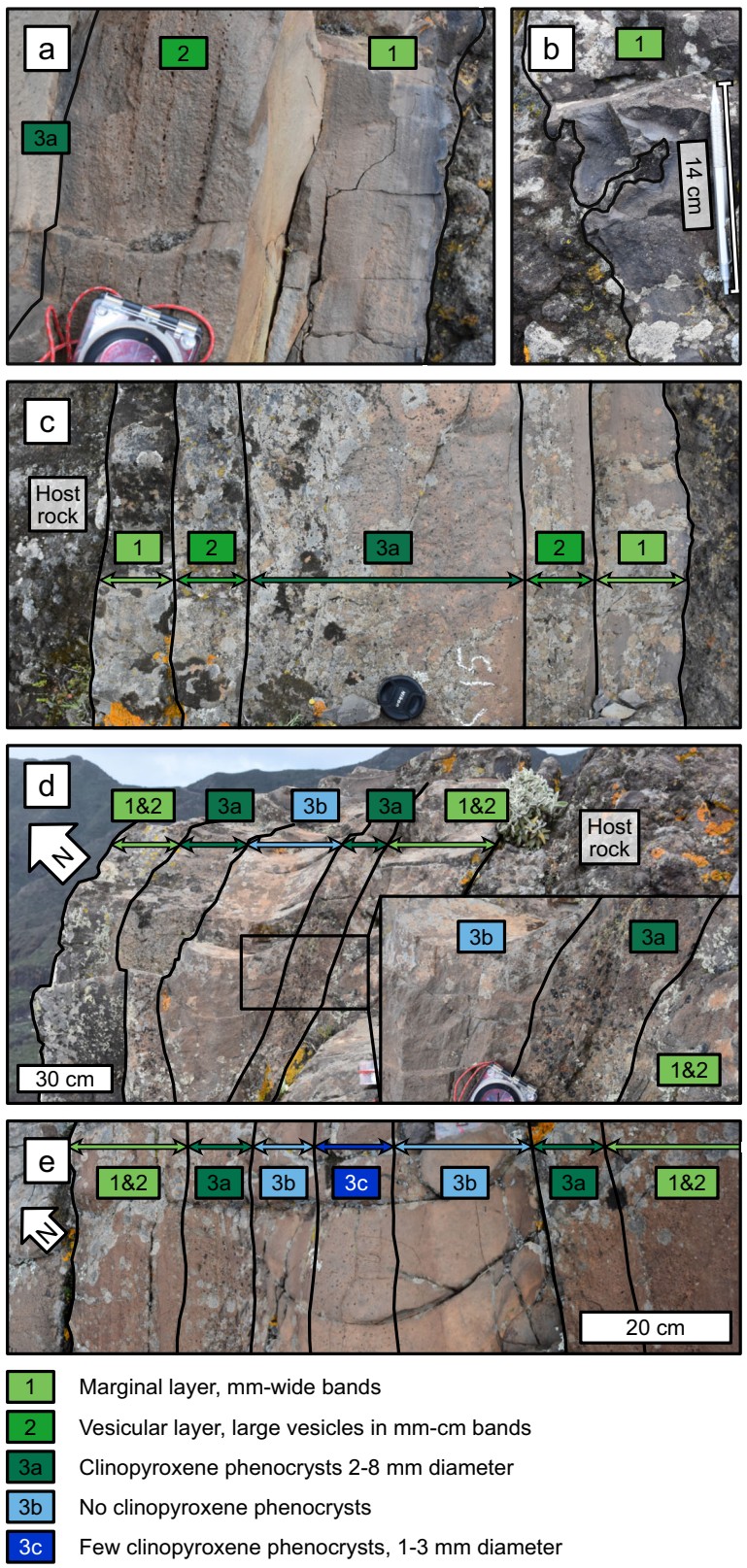

**Fig. 3 | Layers and sub-layers. a** Typical example of the marginal, banded layer and intermediate, vesicular layer found along the studied section of the Carrizales Dyke, with a compass for scale. **b** Example of a highly irregular section of the margin, within metres of the tip of S1. **c** Typical example of the three layers found within S4, with margins that are broadly planar but locally undulous. The dyke is approximately 1 m wide. **d** Sub-layers within the central layer of S4, defined by a variation in the size and concentration of clinopyroxene phenocrysts. **e** The maximum of three sub-layers found within the central layer of S4, defined by clinopyroxene phenocryst content.

Legend:
- **1** Marginal layer, mm-wide bands
- **2** Vesicular layer, large vesicles in mm-cm bands
- **3a** Clinopyroxene phenocrysts 2-8 mm diameter
- **3b** No clinopyroxene phenocrysts
- **3c** Few clinopyroxene phenocrysts, 1-3 mm diameter

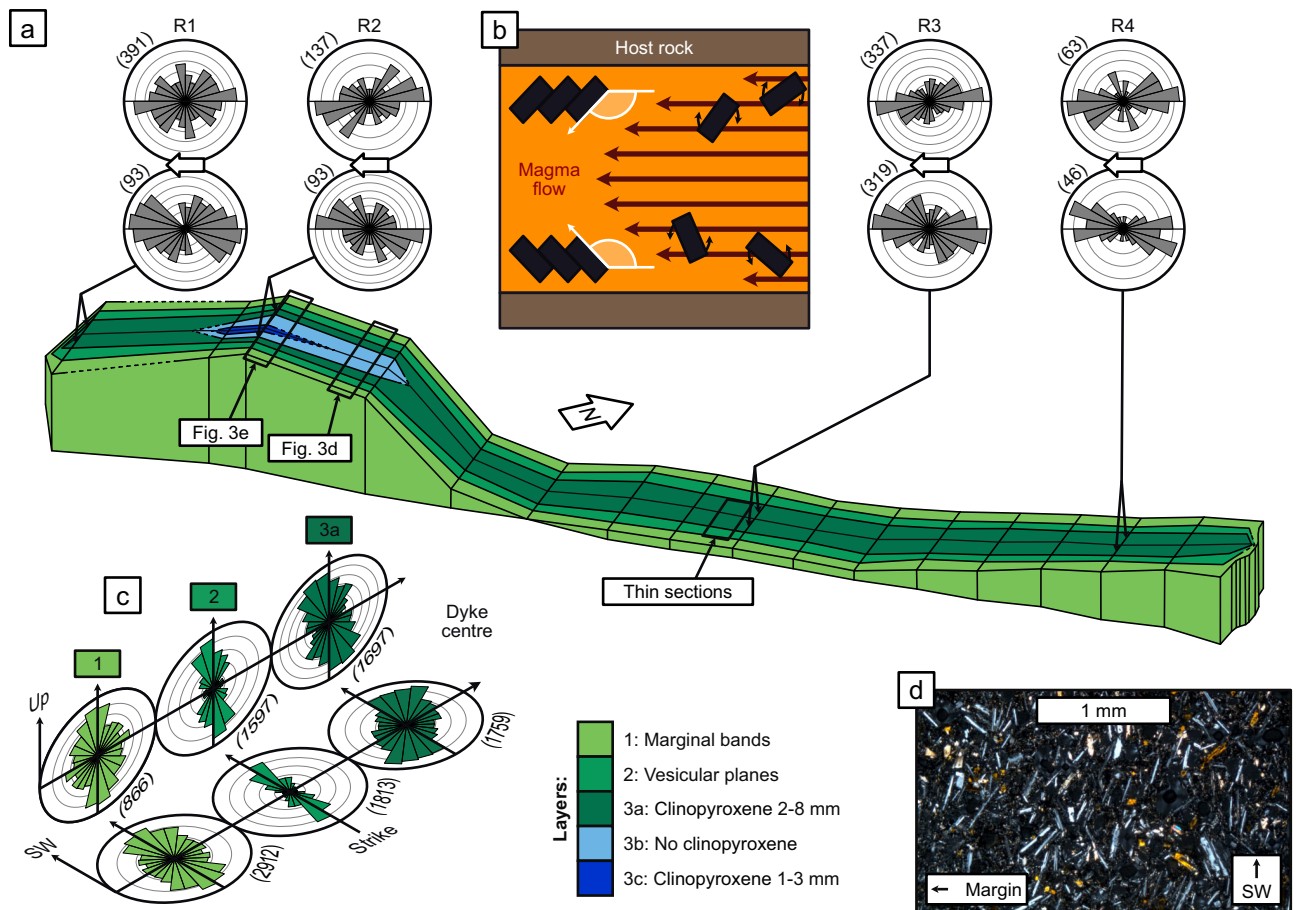

**Fig. 4 | Structure of S4 showing sequence of nested layers and sub-layers.** 3D model of S4 constructed based on measured layer widths along its length, with width dimensions exaggerated by a factor of ten to aid visualisation. **a** Paired rose diagrams from the northern and southern halves of sites R1-4, measuring the orientation of phenocryst long axes from the dyke strike. Values in brackets show the number of crystals outlined to generate each diagram. **b** Schematic cartoon depicting crystal imbrication, with crystals rotating, colliding and stacking to point downstream. **c** Microlite preferred orientations taken from thin sections in vertical and horizontal planes, indicating a SW lateral flow component and a downwards vertical flow component. **d** Example photomicrograph from horizontal plane in marginal layer, in cross-polarised light.

emplacement of the Carrizales Dyke (Fig. 5). The lateral and downwards flow that we identify in each of the three intrusion events is consistent with textural and magnetic studies of other dykes on Tenerife, of a similar age to the Carrizales Dyke, which were used to infer magma propagation away from the main volcanic edifice towards regions of lower elevation[52,53]. We consider it likely that most magma flow was lateral, to the SW, as captured by phenocryst imbrication along the length of S4. Since flow directions are known to show local variation within dyke segments[4], especially near segment edges[29], we infer that the downward component of flow arises because the exposed segments existed at the lower edge of a larger, laterally propagating dyke body. We note that the structure in Fig. 5 is speculative given that so much of the dyke has been eroded or remains concealed; however, we do not aim to be definitive, but to prompt a consideration of the range of segment shapes and the nature of their connections.

Dyke propagation processes can be inferred from the organisation and morphology of segments. Segments that display an en-echelon arrangement, with a consistent orientation and offset direction, are commonly inferred to have formed due to a rotation of the regional stress field in the direction of propagation[3,13]. The Carrizales Dyke segments have inconsistent offset directions, so segmentation has not been driven by the regional stress field. Instead, the dominant control on segment organisation is likely to have been host rock heterogeneities[11], such as the older dyke acting as a barrier between S1 and S2 (Fig. 1f). However, we saw no evidence for faults or lithological

discontinuities controlling segmentation on the studied section of the Carrizales Dyke. Besides the S1-S2 relay, no other relays display large-scale structural controls, which suggests that their triggers for segmentation were small, or have been overwritten by deformation as segments inflated.

Further evidence for propagation processes comes from the dyke margins. Dyke margins are composed of solidified magma from the dyke tip, and as such, they capture dyke-tip processes, at the earliest stage of dyke emplacement. The margins of the Carrizales Dyke are locally irregular, deflecting around clasts in the host rock (Fig. 3a, b), which suggests that the initial fracture was convoluted, rather than having straight edges. This irregularity in fracture width and shape will have caused localised variations in the emplacement rate of the earliest magma, as flow rate is strongly influenced by channel width[1]. Additionally, narrower channels cool faster[1,46–49], so the narrow leading edge of the dyke would have been vulnerable to viscosity increases or solidification. Banded marginal textures similar to those in the Carrizales Dyke have previously been interpreted as evidence of pulsatory propagation driven by thermal instabilities at the leading edge[41]. Therefore, we read the irregular, banded margins as evidence of propagation via stochastic, localised bursts of magma flow along the leading edge of the dyke, analogous to breakouts ahead of a pahoehoe lava flow. This is consistent with our interpretation of S3 and S4 being emplaced in a staggered fashion, with the earlier segment solidifying to the extent that coalescence was inhibited (Fig. 1d).

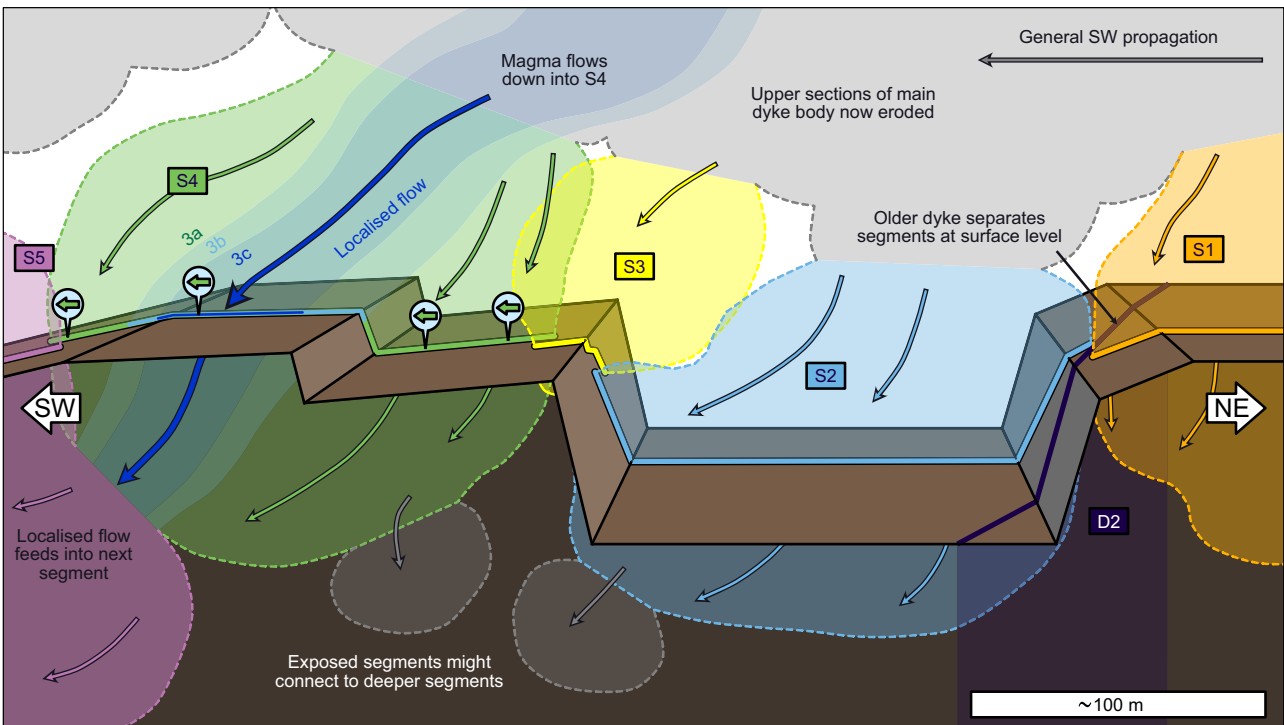

**Fig. 5 | Visualised architecture of the Carrizales Dyke.** Segments connect to upper sections of a larger dyke body (grey), now eroded, as evidenced by the downwards flow within each layer of S4. The exposed segments may connect to others deeper down. Internal sub-layering in S4 suggests that flow localised over time, presumably feeding segments downstream.

## Discussion

We now develop a conceptual model for dyke emplacement, using the Carrizales Dyke as a case study. The first control over dyke architecture comes from the leading edge of the dyke, which is influenced by host rock heterogeneities, regional stresses, and feedback involving cooling and flow. The undulous, banded margins of the Carrizales Dyke suggest that the leading edge was a complex environment, experiencing spatial and temporal variations in magma flow rate. We note that highly localised propagation directions are recorded at the margins of some dykes[4,29], and that pulsatory propagation has been inferred from studies of other dykes[36,41] and sills[54]. Additionally, pulsatory propagation has been seen in analogue experiments of solidifying dykes[19–21], and has been inferred from seismicity during active dyke intrusions[8–10]. So, propagation of the leading edge is likely to be both localised and pulsatory.

Localised, pulsatory propagation is influenced by heterogeneities in host rock properties[55], viscous or elastic instabilities[18,56], and thermal instabilities[19–21], which will exert control over the creation and longevity of dyke segments. For example, regions fed by a wider or faster-flowing magma supply will attain the pressures required to overcome the fracture toughness of the host rock more rapidly, leading to localised bursts of advance, analogous to breakouts on pahoehoe lava flows[57]. Similarly, hotter regions with a thinner outer layer of cooled material may be more likely to rupture, as fluid pressures are transmitted to the host rock more readily. We propose that thermal feedback will amplify the influence of host rock heterogeneities, so that even in the absence of large-scale structural controls such as lithologic boundaries, small-scale deflections of the dyke tip could be enough to split the leading edge into segments, even around individual clasts.

The dynamics within and between established segments at the leading edge influence propagation more broadly. Initially, segments at the leading edge will have limited thickness, making them more vulnerable to cooling and thermal instabilities than the thicker main body of the dyke[1]. However, if segments retain their magma supply, they will inflate. Some segments may coalesce, as in the model of Pollard et al[3].,

but if segments develop a sufficiently thick solid exterior, coalescence will be inhibited, and the segments will likely remain separate, as we see between S3 and S4 (Fig. 1d). The initial catalyst for segmentation might not be obvious, especially once the dyke inflates and deforms the host rock. This was likely the case for the Carrizales Dyke segments, most of which show no obvious structural controls, despite all of them surviving as discrete conduits for three separate intrusion events.

Once segments are established at the leading edge of the dyke, flow will localise within them[21–23], influencing further propagation. Dyke segments are emplaced as broadly linear features, but planar conduits are not thermally efficient, so magma flow will localise into warmer, wider, or faster-flowing regions[1,46–49]. Simple thermal modelling based on the thickness of marginal bands implies that cooling at the dyke tip, leading to viscosity increases and solidification, likely occurs on the timescale of tens of minutes[41]. On the Carrizales Dyke, localisation is evidenced in the nested layers seen in S4 (Figs. 3 and 4). The position and longevity of localised conduits will impact the supply of magma to the leading edge, determining the regions where propagation remains favourable. Only the regions with a steady supply of sufficient volume will attain the pressures required for propagation to continue. The leading edge of the dyke can be envisaged as a series of flow fronts in natural competition, where only the regions with the highest or most stable magma supply can support the advance of the flow front. These dominant segments are more likely to become localised flow pathways, while other areas of the leading edge seal shut, suppressing propagation in that direction. This results in a complex dyke architecture consisting of a main flow pathway surrounded by dead-end branches (Fig. 6).

Our conceptual model for dyke emplacement via a series of connected, lobe-like segments is analogous to the development of pahoehoe lava fields. In lava fields, molten rock is fed from a source region and travels via stochastic breakouts at the flow front, forming a connected lava 'pad' that experiences progressive localisation towards lava tubes[57,58]. Our inferred dyke structure is superficially similar to

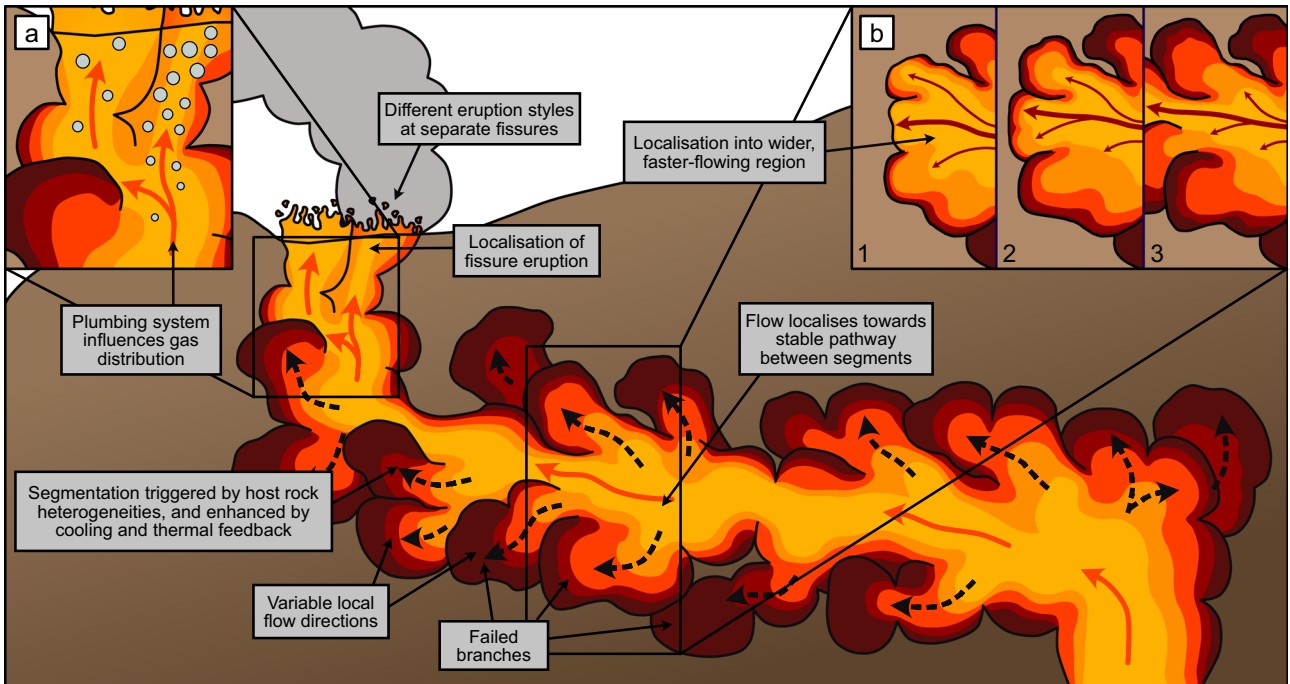

**Fig. 6 | Schematic cartoon of complex dyke architecture.** Propagation occurs via the sequential emplacement of lobes at the leading edge, where host rock heterogeneities drive segmentation, followed by flow localisation enhanced by thermal feedback (inset **b**). The result is a branching, convoluted structure, which then impacts the transport of magma and volatiles. Branching pathways beneath fissures (inset **a**), could cause gas segregation, leading to different eruption styles at neighbouring fissures in the same eruption.

analogue experiments with solidifying fluids[19–21]. Parallels can also be drawn with sills in sedimentary basins, which have been shown through seismic reflection surveys to have a segmented structure and host flow pathways on a larger scale[14–16].

Understanding segmentation is crucial for understanding dyke emplacement dynamics, and for estimating the likelihood, location, and behaviour of a potential basaltic fissure eruption, which together comprise the most vital pieces of information for hazard mitigation. The migration of seismicity during active dyke emplacement provides some insight into the evolving architecture of the plumbing system, with previous studies suggesting that propagation may occur on multiple fronts, at different rates, involving stalling and restarting[8–10]. Our interpretation of the Carrizales Dyke supports this proposition, and demonstrates the potential for future interpretations of seismic datasets to illuminate the dynamics of segment formation, allowing better forecasts of eruption location and likelihood. A computational model encompassing the combined effects of flow dynamics, rock mechanics, and thermal feedback may be a long way off, but a conceptual model that can relate segment formation dynamics inferred from seismic evidence to potential eruption behaviour would be invaluable to hazard mitigation efforts.

Complex dyke architecture will likely influence the longevity and behaviour of fissure eruptions. The time taken for a conduit to seal shut is heavily dependent on its width and flow rate[1], which will be influenced by localisation dynamics during the propagation phase[46–49]. Convoluted, branching pathways may impact the way that magma degasses by controlling the transport of bubbles and gas slugs[2,27], which, in turn, could influence eruption style[59]. This could potentially lead to different vents being fed by magma of different gas contents, leading to distinct eruption styles[28]. Furthermore, complex degassing pathways and the suppression of large-scale vertical convection may create buoyancy contrasts that favour local downwards propagation, potentially encouraging long-distance lateral transport.

Our conceptual model also has implications for interpreting dykes in the field. Firstly, when interpreting segments exposed on one plane,

it should be appreciated that they may be limited in their vertical and horizontal extents, with a complex rather than simple connection to a continuous parent body. Within segments, flow directions may have varied spatially and temporally due to flow localisation[22,23,44], and this should inform the collection of samples and the interpretation of rock fabrics, as flow direction may depend on the position of a sample relative to the ends of a segment and across the width of the dyke[4,29,39].

With our model of dyke emplacement, we seek to challenge the common oversimplification of dykes to a single, planar structure, and the neglect of thermal feedback in guiding propagation. Currently, only a few numerical[60] and analogue[19–21,44,61] models have attempted to investigate solidification and propagation simultaneously. Our interpretation of the Carrizales Dyke suggests that flow localisation during the propagation phase is linked intrinsically to segment formation, and further research could address this via numerical or analogue modelling, comparing against interpretations of seismicity during active dyke emplacement.

## Methods
We use evidence from a range of scales to infer the emplacement history of the Carrizales Dyke, combining field evidence of segment morphology with evidence of internal layered textures, and with small-scale textural evidence of magma flow directions. This allows us to infer how magma flow evolved within the studied dyke system.

### Segment geometry and rock textures
At the largest scale, we mapped five segments in the study area and measured their lengths in satellite images (Fig. 1). The widths of dyke layers were measured manually, and used to construct the 3D model in Fig. 4. Only S4 is accessible along its entire length, so this was where we focussed our efforts regarding rock textures; other segments are only accessible at one location along their length, although they are visible on the steep slopes. We took photographs parallel to the plane of the dyke exposure, then used these photos to outline clinopyroxene phenocrysts in the central layer, which was done manually using ImageJ[62].

## Thin sections

Oriented samples were taken from each layer at a site midway along S4 (location in Supplementary Fig. 1), and thin sections were cut on horizontal and vertical planes, perpendicular to the dyke margin. Thin section images were thresholded in ImageJ[62] to capture the outlines of microlites, mostly plagioclase, against the dark groundmass. The cross-sectional shapes of phenocrysts and microlites capture the preferred orientation of crystals, from which we infer the direction of magma flow.

## Data availability

Data for precise location of Carrizales Dyke, segment lengths, layer widths and crystal orientations are available in the Supplementary Information file.

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

## Acknowledgements

This research was funded by the UK's Natural Environment Research Council (NE/S007431/1, awarded to C.A.). We would like to thank Alexis Schwartz for providing fieldwork support.

## Author contributions

C.A. contributed conceptualisation of the study, data collection in the field, data analysis and interpretation, figure creation, and writing the manuscript. E.W.L. contributed conceptualisation of the study, data interpretation, and writing the manuscript. R.J.B. and A.L. contributed to data collection and interpretation of dyke morphology in the field.

## Competing interests

The authors declare no competing interests.
