## [Transparent Peer Review file · Nature Communications]

Magma flow localisation during dyke propagation produces complex magma transport pathways

Corresponding Author: Dr Ceri Allgood

Version 0:

Reviewer comments:

Reviewer #1

(Remarks to the Author)

Review of manuscript entitled "Localisation during dyke propagation produces complex magma transport pathways", by Allgood et al.

I read this manuscript with great interest. It deals with a detailed description of the 3-dimensional structure of a well-exposed dyke in Tenerife. Exposure in the study area allowed the authors to map in great details the complex shape of the dyke, made of individual segments that may be connected in the third dimension, as well as the internal structure of the dyke. These latter observations allow the authors to constrain the complex magma flow distribution within a dyke segment. In particular, the authors show that magma flow became localised through a channel of limited extent compared to the length of the dyke segment.

Overall, I found the manuscript very well written, and the graphic design of the figures is of very high quality. The conclusions are very interesting, and I think relevant. I like the use of a detailed case study used to reveal a fundamental process, like the authors do in this manuscript.

Nevertheless, I would have some criticisms and concerns with respect to this study. My main concerns is that the text overall lacks support, and some paragraphs appear like a list of statements without reference to figures or literature (I come back to this in details below). Therefore, the study is written as a very good story, which sometimes lack scientific ground. As it is, the manuscript is not suitable for publication in Nature Communications. Below I provide detailed comments to support my conclusions and help the authors improving their nice study.

Title

The title can be misleading. What do the authors mean by "Localisation"? It could be interpreted by tectonics structural geologists as deformation localisation. I invite the authors to specify "Magma flow localisation ...".

Introduction

The introduction is very interesting and sets the stage of the study. Nevertheless, I think it lacks some important literature. In the second paragraph, the authors introduce the segmented nature of dykes, and introduce the relevance of constraining the 3-dimensional shapes of the dykes. I invite the authors to refer to the large literature on 3-dimensional structure of sills, which are sheet intrusions made of segments, very similar to the structure described by the authors. Such literature includes, but is not restricted to, 3D seismic imaging of sills (Thomson and Hutton, 2004; Hansen and Cartwright, 2006; Schofield et al., 2012b; Magee et al., 2016; Schmiedel et al., 2017; Magee et al., 2019), field observations of sill segments (Nicholson and Pollard, 1985; Hutton, 2009; Schofield et al., 2012a; Spacapan et al., 2017; Galland et al., 2019).

Also some literature of the segmented nature of dykes and the associated complex magma flow could be added, either field-based studies (Poland et al., 2004; Schmiedel et al., 2021; Stephens et al., 2021) of analogue modelling studies (Takada, 1990; Kavanagh et al., 2018). This latter reference is particularly relevant, as Kavanagh et al. (2018) image a complex, localised analogue magma flow within a single dyke segment, i.e. similar to what the authors observe.

I thus invite the authors to better place the results of their study in the scientific context.

Finally, the third chapter of the introduction is rather speculative and vague. The authors introduce numerous potential implications of revealing the 3D structures of dykes and associated complex magma flow, but this paragraph contains only one citation, the rest being a list of statements not supported by the literature.

In general, the authors are honest by writing that they present a conceptual model. Nevertheless, this conceptual model includes magma flow dynamics, rock mechanics for dyke propagation, and thermal feedback, i.e. quite a complex set of processes that interplay. However, the data provided by the authors only support the magma flow dynamics, but the authors provide little evidence of thermal effects, and no structural data to discuss rock mechanics associated with dyke propagation. As such, this conceptual model is very speculative.

Section "Intrusion events"

Lines 141-142. The authors write " There is a chilled margin between the intermediate and central layers, with a noticeable drop in grain size." This statement is not supported by data. Please provide data that supports this statement.

Discussion

Section "Segmentation and stochastic propagation"

This section, and in particular the second paragraph, is quite speculative. Except the irregular shape of the dyke segments, the authors do not provide structural elements (fractures, faults, etc) that would feed this discussion. The second paragraph introduces and links several concepts and mechanisms, with only one citation to the literature (a paper of the authors themselves) and no reference to any data and figure in this manuscript. I find it too conceptual and speculative, and not enough supported by data and/or literature.

Section "Thermal feedback"

This section discusses the potential thermal effects of magma flow localisation. Again, I find this discussion too qualitative and not enough supported by the literature or data. One can also note that the authors have not conducted any thermal calculations, even in orders of magnitude, to support their discussion.

The main argument of the authors supporting thermal effects is the observation of magma flow localisation. However, Kavanagh et al. (2018) show that localised magma flow can occur without thermal effects. Thus, the authors need more observations/data/analyses to feed this discussion.

Section "Conceptual model for dyke emplacement"

This section is probably the most problematic. It develops a conceptual model of the potential processes that may be at work at the leading edge of the studied dyke. The problem here is that there is no observation of the leading edge of the dyke, therefore this whole discussion is poorly linked to the study (very few references to the data), and can almost be written as a general, simplified review of potential processes at the leading edge of a dyke. Thus, the data provided in this study barely support this discussion, which appears somehow off-subject. In addition, the third and fourth paragraphs of this section contain almost no reference to the literature (only one) and references to qualitative observations. These paragraphs appear as a list of statements, which provide the feeling to the reader that they are quite speculative. This is not sufficient for publication in Nature Communications.

All in all, I feel that this study is very interesting, the figures are very elegant, the data presented are very sound, but the ambitious conclusions and conceptual model are not supported by the data. This makes the study, as it is, not suitable for publication in Nature Communications. I thus recommend the authors to considerably strengthen their data set, or to reduce the ambition of the conclusions of the study.

I hope these comments will help the authors to strengthen their manuscript.

Best.

References

- Galland, O., Spacapan, J.B., Rabbal, O., Mair, K., Soto, F.G., Eiken, T., Schiuma, M., Leanza, H.A., 2019. Structure, emplacement mechanism and magma-flow significance of igneous fingers – Implications for sill emplacement in sedimentary basins. *J. Struct. Geol.* 124, 120–135.
- Hansen, D.M., Cartwright, J.A., 2006. Saucer-shaped sill with lobate morphology revealed by 3D seismic data: implications for resolving a shallow-level sill emplacement mechanism. *J. Geol. Soc. London* 163, 509-523.
- Hutton, D.H.W., 2009. Insights into magmatism in volcanic margins: bridge structures and a new mechanism of basic sill emplacement, Theron Mountains, Antarctica. *Petroleum Geoscience* 15, 269-278.
- Kavanagh, J.L., Burns, A.J., Hilmi Hazim, S., Wood, E.P., Martin, S.A., Hignett, S., Dennis, D.J.C., 2018. Challenging dyke ascent models using novel laboratory experiments: Implications for reinterpreting evidence of magma ascent and volcanism. *J. Volcanol. Geotherm. Res.* 354, 87-101.
- Magee, C., Muirhead, J.D., Karvelas, A., Holford, S.P., Jackson, C.A.L., Bastow, I.D., Schofield, N., Stevenson, C.T.E., McLean, C., McCarthy, W., Shtukert, O., 2016. Lateral magma flow in mafic sill complexes. *Geosphere* 12, 809-841.
- Magee, C., Muirhead, J.D., Schofield, N., Walker, R.J., Galland, O., Holford, S., Spacapan, J.B., Jackson, C.A.L., McCarthy, W., 2019. Structural signatures of igneous sheet intrusion propagation. *J. Struct. Geol.* 125, 148-154.
- Nicholson, R., Pollard, D.D., 1985. Dilation and linkage of echelon cracks. *J. Struct. Geol.* 7, 583-590.
- Poland, M.P., Fink, J.H., Tauxe, L., 2004. Patterns of magma flow in segmented silicic dikes at Summer Coon volcano, Colorado: AMS and thin section analysis. *Earth Planet. Sci. Lett.* 219, 155-169.
- Schmiedel, T., Burchardt, S., Mattsson, T., Guldstrand, F., Galland, O., Palma, J.O., Skogby, H., 2021. Emplacement and Segment Geometry of Large, High-Viscosity Magmatic Sheets. *Minerals* 11.
- Schmiedel, T., Kjöberg, S., Planke, S., Magee, C., Galland, O., Schofield, N., Jackson, C.A.-L., Jerram, D.A., 2017. Mechanisms of overburden deformation associated with the emplacement of the Tulipan sill, mid-Norwegian margin. *Interpretation* 5, SK23-SK38.
- Schofield, N., Brown, D.J., Magee, C., Stevenson, C.T., 2012a. Sill morphology and comparison of brittle and non-brittle

emplacement mechanisms. *J. Geol. Soc.* 169, 127-141.

Schofield, N., Heaton, L., Holford, S.P., Archer, S.G., Jackson, C.A.L., Jolley, D.W., 2012b. Seismic imaging of "broken bridges": linking seismic to outcrop-scale investigations of intrusive magma lobes. *J. Geol. Soc.* 169, 421-426.

Spacapan, J.B., Galland, O., Leanza, H.A., Planke, S., 2017. Igneous sill and finger emplacement mechanism in shale-dominated formations: a field study at Cuesta del Chihuido, Neuquén Basin, Argentina. *J. Geol. Soc.* 174, 422-433.

Stephens, T.L., Walker, R.J., Healy, D., Bubeck, A., 2021. Segment tip geometry of sheet intrusions, II: Field observations of tip geometries and a model for evolving emplacement mechanisms. *Volcanica* 4, 203 - 225.

Takada, A., 1990. Experimental study on propagation of liquid-filled crack in gelatin: shape and velocity in hydrostatic stress condition. *J. Geophys. Res.* 95, 8471-8481.

Thomson, K., Hutton, D., 2004. Geometry and growth of sill complexes: insights using 3D seismic from the North Rockall Trough. *Bull. Volcanol.* 66, 364-375.

Reviewer #2

(Remarks to the Author)

In their manuscript "Localisation during dyke propagation produces complex magma transport pathways", Allgood et al. present data on the external and internal architecture of a basaltic dyke exposed in the Teno massif of Tenerife, Canary Islands, Spain. They used field observations, image analysis and thin section analysis to derive information on magma flow during dyke emplacement. They then use this data to infer a model of dyke propagation that resembles the development of lava flow fronts, including lobes and channelized flow. The study presented in the manuscript relies on quantitative data and comes up with a novel view on dyke propagation, which should be of interest to the readers of *Nature Communications*. The manuscript is well written and clear. However, there are a few major points that need to be considered in a revised version of the manuscript. All my comments are listed below, but the major ones to focus on are clearly 2, 5/6, 7, 12, 14, 15 and 16. More specifically, (a) I miss quantitative data on dyke segment geometry and pre-existing structures. (b) I miss information on the host rock and minerals. (c) I want to see discussion of alternative explanations regarding flow localisation (other than thermal effects) and downward magma flow. Moreover, I suggest adding a wider selection of references and gave some examples below.

Kind regards,
Steffi Burchardt

List of suggested improvements

1. Title: I suggest adding "flow" to the title to read "flow localization during dyke propagation" as just "localisation" is too unspecific.
2. Line 45: the Tajogaite eruption on La Palma occurred in 2021 not 2022.
3. More information on the host rock is needed. The authors mention that the host rock is volcanoclastic and infer that it was potentially unconsolidated which affected dyke propagation. However, the rock type is not described in enough detail. Also, from the field photographs it appears that the host rock varies from massive (e.g. Suppl. Fig. 2a) to layered (e.g. Suppl. Fig. 2b). Some researchers studying magmatic sheet propagation are of the strong opinion that the host rock has a considerable effect on sheets (e.g. Schmiedel et al., 2017; Vachon & Hieronymus, 2017; Galland et al., 2019), whereas others think it the host rock is secondary to other parameters (e.g. Schmiedel et al., 2021; Stephens et al., 2021). Also, there is literature on dyke propagation through unconsolidated material that should be discussed and cited (Petronis et al., 2013).
4. Fig. 1: Photographs are too small/dark. While I appreciate that the photos are available in a larger format in the supplementary, I think most readers will not go there. The photographs in Fig. 1 (c-e) are too small, the lines are hard to see (especially the green and blue ones), and e is far too dark. I don't think *Nature Communication* restricts the size of figures so much. Also, I would prefer to have an inset with the location data in Fig. 1 instead of the supplementary.
5. Dyke morphology: the morphology of magmatic sheet intrusions has been used extensively to derive information on sheet propagation. Since, the Carrizales dyke is so well exposed, it would be useful to present this data (segment length vs. thickness vs. tip geometry) and put it into a scientific context (see Schmiedel et al., 2021). The authors mention in the Methods section that segment lengths and thicknesses were measured, but apart from Fig. 1 and an average thickness, this data is neither presented nor discussed. The thickness along the segment would be particularly interesting for segment 4, where it should be related to the emplacement of the lenticular layers 3b and 3c.
6. Three-dimensional architecture: The study of the 3D architecture of the Carrizales dyke would have been greatly aided by photogrammetry and 3D visualisation (e.g. Schmiedel et al., 2021). Also, quantitative data on segment geometries, overlap etc. such as in Delaney et al. (1986) should be presented and discussed. Overall, the section "Three-dimensional architecture" is not really what it promises. Instead, it describes the relationship between segments in a qualitative way. Additionally, since there is a later section called "inferred architecture" the naming of the sections becomes confusing. I suggest to rename the "Three-dimensional architecture" section to something like "segment geometry."
7. Flow directions: It would be great to add an example photograph of the imbricated phenocrysts and of the imbricated microlites in thin section to Fig. 4. Please also add the microstructural images to the Supplementary information. Also, what minerals were considered? What type of crystal habitus do they have? Crystal shapes (whether needly or platy) and phenocryst arrangements (single crystals or glomerocrysts) strongly affect the alignment. For instance, magma flow in the Sandfell laccolith in Iceland is outlined (in part) by the alignment of single, platy plagioclase phenocrysts (Mattsson et al., 2018). Likewise, needly hornblende phenocrysts align in the Cerro Bayo cryptodome, but plagioclase phenocrysts do not (Burchardt et al., 2019). The latter because plagioclase occurs as glomerocrysts.
8. Figure 3: Please add a legend instead of describing the layers in the figure caption. Also consider add a close-up photo of each layer.
9. Figure 4 and Suppl. Figs. 4a-d and Suppl. Fig. 5. Each rose diagram needs to be labelled with the amount of data

displayed.

10. Suppl. Fig. 3a. What is the yellow dashed line? (I guess it is the dyke margin as mentioned in the caption to Suppl. Fig. 3b. But please also add to 3a.

11. Segmentation and stochastic propagation: “Stochastic” is not a term that has been used in the intrusion community so far. Please introduce it with a definition.

12. Thermal feedback: Analogue models of dyke propagation by Janine Kavanagh and her group show flow localisation due to fluid dynamics within the intrusion and not related to thermal effects (e.g. Williams et al., 2022; Chalk and Kavanagh, 2024). While actual dykes of course must experience the effect of magma temperature, thermal feedback might not be the only mechanism responsible for flow localisation. I therefore suggest to discuss alternatives, such as fluid dynamics and even shear thinning rheology of magma.

13. Conceptual model, Line 239ff: “We note that highly localised propagation is recorded at the margins of some dykes⁴, and that pulsatory propagation has been inferred from studies of other intrusions^{25,30}.” This sentence sounds to me like “other intrusions” should be intrusions that are not dykes. However, both citations refer to dyke studies. Pulsatory magma emplacement seems to be the norm these days and has been recognised in sills (e.g. Köpping et al., 2022), laccoliths (e.g. Witcher et al., 2024) and plutons (e.g. Menand et al., 2011). Also, I would suggest to link back to the seismic monitoring of dyke propagation here, with the Holuhraun dyke as a prime example of a dyke that propagated in segments (Sigmundsson et al., 2015).

14. Downward magma flow: The data presented in this manuscript are interpreted indicating downward flow of magma. However, fabrics in solidified dykes are not recording primary, i.e. emplacement-related, magma movement. Since the propagation stage is usually followed by hours to weeks of magma flow through a fissure before flow localises to vents in fissure eruptions, other processes can affect magma flow directions. For instance, dyke closure, sagging, and consolidation may alter the fabrics that will be preserved in the rock (see Andersson et al., 2016 Discussion and Fig. 8). Sagging may explain the downward flow indicators in the Carrizales dyke and should be discussed, since the conceptual model relies on the downward flow to be primary.

15. Magma flow studies on dykes in similar settings: The authors should discuss their findings in the context of other studies of magma flow in dykes, particularly in volcanic rift zones. Several papers focus on fabrics that indicate magma flow in Canary Island dykes, e.g. Delcamp et al. (2015). In this paper, the authors discuss the effect of topography causing downward flow along the rift zone.

16. Pre-existing structures: The Teno massif is the remnant of a volcanic rift zone. Dyke emplacement there was therefore strongly controlled by tectonic stresses, topography, and pre-existing structures, such as faults (cf. Delcamp et al., 2012 on the Anaga massif). Dykes tend to follow pre-existing weaknesses that have a favourable orientation in the prevailing stress field at the time of emplacement. Also, flow localisation is more likely where several fractures intersect. Therefore, it is important to provide data on the orientation of the dyke and other structures in the vicinity (e.g. Greiner et al., 2023). The only other structure that is mentioned is D2. But can the existence and influence of fractures really be ignored? The photos in Suppl. Figs. 2b and c show fractures sub-parallel to the dykes.

Andersson, M., Almqvist, B. S., Burchardt, S., Troll, V. R., Malehmir, A., Snowball, I., & Kübler, L. (2016). Magma transport in sheet intrusions of the Alnö carbonatite complex, central Sweden. *Scientific Reports*, 6(1), 27635.

Burchardt, S., Mattsson, T., Palma, J. O., Galland, O., Almqvist, B., Mair, K., ... & Sun, Y. (2019). Progressive growth of the Cerro Bayo cryptodome, Chachahuén volcano, Argentina—Implications for viscous magma emplacement. *Journal of Geophysical Research: Solid Earth*, 124(8), 7934-7961.

Chalk, C. M., & Kavanagh, J. L. (2024). Up, down, and round again: The circulating flow dynamics of flux-driven fractures. *Physics of Fluids*, 36(3).

Delcamp, A., Petronis, M. S., & Troll, V. R. (2015). Discerning magmatic flow patterns in shallow-level basaltic dykes from the NE rift zone of Tenerife, Spain, using the Anisotropy of Magnetic Susceptibility (AMS) technique. *Geological Society, London, Special Publications*, 396(1), 87-106.

Galland, O., Spacapan, J. B., Rabbal, O., Mair, K., Soto, F. G., Eiken, T., ... & Leanza, H. A. (2019). Structure, emplacement mechanism and magma-flow significance of igneous fingers—Implications for sill emplacement in sedimentary basins. *Journal of Structural Geology*, 124, 120-135.

Greiner, S. H., Burchardt, S., Sigmundsson, F., Óskarsson, B. V., Galland, O., Geirsson, H., & Rhodes, E. (2023). Interaction between propagating basaltic dikes and pre-existing fractures: A case study in hyaloclastite from Dyrfjöll, Iceland. *Journal of Volcanology and Geothermal Research*, 442, 107891.

Köpping, J., Magee, C., Cruden, A. R., Jackson, C. A. L., & Norcliffe, J. R. (2022). The building blocks of igneous sheet intrusions: Insights from 3-D seismic reflection data. *Geosphere*, 18(1), 156-182.

Mattsson, T., Burchardt, S., Almqvist, B. S., & Ronchin, E. (2018). Syn-emplacement fracturing in the Sandfell laccolith, eastern Iceland—Implications for rhyolite intrusion growth and volcanic hazards. *Frontiers in Earth Science*, 6, 5.

Menand, T., de Saint-Blanquat, M., & Annen, C. (2011). Emplacement of magma pulses and growth of magma bodies. *Tectonophysics*, 500(1-4), 1-2.

Petronis, M. S., Delcamp, A., & Van Wyk De Vries, B. (2013). Magma emplacement into the Lemptégy scoria cone (Chaîne Des Puys, France) explored with structural, anisotropy of magnetic susceptibility, and Paleomagnetic data. *Bulletin of volcanology*, 75, 1-22.

Schmiedel, T., Galland, O., & Breitzkreuz, C. (2017). Dynamics of sill and laccolith emplacement in the brittle crust: role of host rock strength and deformation mode. *Journal of Geophysical Research: Solid Earth*, 122(11), 8860-8871.

Schmiedel, T., Burchardt, S., Mattsson, T., Guldstrand, F., Galland, O., Palma, J. O., & Skogby, H. (2021). Emplacement and segment geometry of large, high-viscosity magmatic sheets. *Minerals*, 11(10), 1113.

Sigmundsson, F., Hooper, A., Hreinsdóttir, S., Vogfjörð, K. S., Ófeigsson, B. G., Heimisson, E. R., ... & Eibl, E. P. (2015). Segmented lateral dyke growth in a rifting event at Bárðarbunga volcanic system, Iceland. *Nature*, 517(7533), 191-195.

Stephens, T., Walker, R., Healy, D., & Bubeck, A. (2021). Segment tip geometry of sheet intrusions, II: Field observations of tip geometries and a model for evolving emplacement mechanisms. *Volcanica*, 4(2), 203-225.

Vachon, R., & Hieronymus, C. F. (2017). Effect of host-rock rheology on dyke shape, thickness and magma overpressure.

Geophysical Journal International, 208(3), 1414-1429.

Williams, K. M., Kavanagh, J. L., & Dennis, D. J. C. (2022). Focused flow during the formation and propagation of sills: Insights from analogue experiments. *Earth and Planetary Science Letters*, 584, 117492.

Witcher, T., Burchardt, S., Mattsson, T., Heap, M. J., & McCarthy, W. (2024). Development of permeable networks by viscous-brittle deformation in a shallow rhyolite intrusion. Part 1: Field evidence. *Journal of Volcanology and Geothermal Research*, 454, 108166.

Version 1:

Reviewer comments:

Reviewer #2

(Remarks to the Author)

This is a re-review of the manuscript NCOMMS-24-81975 by Allgood et al. now titled "Magma flow localisation during dyke propagation produces complex magma transport pathways." I have been asked to review how the authors addressed both my and the comments of Reviewer 1.

Overall, I note that Reviewer 1 and I had many overlapping comments.

Reviewer 1

- Introduction: R1 requested to revise the introduction and include references on sill structure and dyke segmentation. This is now done in the revised manuscript. R1 also asked the authors to clarify the implications of their study and add references. This is done as well.
- Conceptual model: R1 was concerned that the data only allowed inferences based on data that can be interpreted in terms of magma flow dynamics, and not – as originally claimed – rock mechanics and thermal feedback. The authors disagree and have now clarified how their data also supports interpretation on dyke tips processes, including magma-host interaction. The authors have added to their description of the dyke margins and to Fig. 3. To support this.
- R1 requested additional data on chilled margins between layers. Done.

Shared comments

- Title: The revised manuscript title correctly reflects the content.
- Additional data on structures. The authors quickly describe that there are no major structures like fractures or faults they see the dyke intersection with and that one cannot be sure about the age relationship between the dyke and fractures. This only partly answers the question. Measurements of the few fractures (and other dykes) would show whether there is a preferred orientation of fractures and dykes in the study area, which could provide information on the influence of the stress field and potentially pre-existing weaknesses the dyke uses (and now covers). I appreciate that this data is hard to obtain afterwards in case the authors did not collect it in the first place. However, the revised description of the scarcity of fractures is sufficient.
- Discussion: Thermal effects: R1 found this section too speculative, and I asked to discuss alternatives. In the revised manuscript, the authors now reference modelling studies on thermal effects.
- Conceptual model: R1 found the entire model description too speculative, not linked with either the data or the literature, and therefore "off-topic". The authors disagree, as their data on dyke margins, segments and flow localisation provide insight into dyke tip processes. I also had some requests to clarify this part of the manuscript, e.g. regarding the pulsatory magma emplacement and by linking back to seismic data. They have rewritten this part of the discussion and included more references to their own data and the literature. As such, I find their revisions satisfactory.
- More references in general. Done in a satisfactory way.

My (Reviewer 2) comments

- Lack of quantitative data on dyke segment geometry: This data is now available in the manuscript and the supplementary files.
- Figure quality: fixed.
- More information on the host rock and minerals. Mostly done. However, I suggested to cite and discuss literature on dyke propagation through unconsolidated material (Petronis et al., 2013), which the authors did not take on board, because one cannot know the state of the host rock at the time of dyke propagation. Yet, they state that the dyke "intruded when the volcanoclastic host rock was relatively unconsolidated." Also, a lot of the interpretation relies on the interaction of the dyke tip with the irregularities in the host rock. So I encourage the authors to address this comment.
- Flow directions: photographs and microphotographs are now provided. Thank you.
- Discussion of alternative explanations regarding downward magma flow. This is done in a satisfactory way.
- Magma flow studies on dykes in similar settings: This is now done (lines 214 ff).

So to conclude, I think the authors did a very good job addressing most of our comments. The manuscript had improved substantially, especially with regards to better backing up interpretation with data and literature references.

I therefore suggest to accept the manuscript for publication.

Kind regards,

Steffi Burchardt

General remarks for both reviewers:

We thank the reviewers and editor for their very helpful comments. We have made changes in response to every point made and believe that this has substantially improved the manuscript. In our point-by-point response below, changes made are highlighted in red for your convenience, whereas responses to the reviewers' comments are in blue.

Please note that the manuscript has been restructured to comply with Nature Communications formatting. The order of information remains mostly unchanged, except for the Methods section, which has been moved to the end of the manuscript, but some headings and sub-headings have been removed or adapted to meet the journal requirements. All numbering has been removed from section headings.

The structural changes are as follows:

- Section 2 (“The Carrizales Dyke”), which served to introduce the locality of the study, has now been incorporated into the Results section.
- We have updated the sub-headings in the Results sections to more accurately reflect the content, pairing each type of evidence with its interpretation:
 - “Three dimensional architecture” is now “Segment geometry and inferred propagation directions”
 - “Intrusion events” is now “Textural layering and intrusion events”
 - “Flow localisation” is now “Sub-layering and flow localisation”
 - “Flow directions” is now “Crystal imbrication and flow directions”
- The main points from the original “Thermal feedback” sub-section are now included in “Sub-layering and flow localisation”, where other potential flow localisation mechanisms are discussed.
- The original “Inferred architecture” sub-section has been renamed “Inferred architecture and emplacement” and moved into the Results section. This now includes the important points from the original “Segmentation and stochastic propagation” sub-section.
- Section 6 (“Implications and Conclusions”) has been incorporated into the new Discussion section. All sub-headings have been removed from the Discussion section to comply with journal requirements.
- The Methods section has been moved to the end of the manuscript.
- Figures have been removed from the main text and captions have been relocated to the end of the manuscript.

Figures have been resized and optimised to meet journal requirements.

Response to reviewer 1:

Review of manuscript entitled "Localisation during dyke propagation produces complex magma transport pathways", by Allgood et al.

I read this manuscript with great interest. It deals with a detailed description of the 3-dimensional structure of a well-exposed dyke in Tenerife. Exposure in the study area allowed the authors to map in great details the complex shape of the dyke, made of individual segments that may be connected in the third dimension, as well as the internal structure of the dyke. These latter observations allow the authors to constrain the complex magma flow distribution within a dyke segment. In particular, the authors show that magma flow became localised through a channel of limited extent compared to the length of the dyke segment.

Overall, I found the manuscript very well written, and the graphic design of the figures is of very high quality. The conclusions are very interesting, and I think relevant. I like the use of a detailed case study used to reveal a fundamental process, like the authors do in this manuscript.

Nevertheless, I would have some criticisms and concerns with respect to this study. My main concern is that the text overall lacks support, and some paragraphs appear like a list of statements without reference to figures or literature (I come back to this in details below). Therefore, the study is written as a very good story, which sometimes lack scientific ground. As it is, the manuscript is not suitable for publication in Nature Communications. Below I provide detailed comments to support my conclusions and help the authors improving their nice study.

Title

The title can be misleading. What do the authors mean by "Localisation"? It could be interpreted by tectonics structural geologists as deformation localisation. I invite the authors to specify "Magma flow localisation ...".

A similar point was made by Reviewer 2. We agree and we have changed the title to specify magma flow localisation, to avoid any confusion. Thank you for the suggestion.

Introduction

The introduction is very interesting and sets the stage of the study. Nevertheless, I think it lacks some important literature.

In the second paragraph, the authors introduce the segmented nature of dykes, and introduce the relevance of constraining the 3-dimensional shapes of the dykes. I invite the authors to refer to the large literature on 3-dimensional structure of sills, which are sheet intrusions made of segments, very similar to the structure described by the authors. Such literature includes, but is not restricted to, 3D seismic imaging of sills (Thomson and Hutton, 2004; Hansen and Cartwright, 2006; Schofield et al., 2012b; Magee et al., 2016; Schmiedel et al., 2017; Magee et al., 2019), field observations of sill segments (Nicholson and Pollard, 1985; Hutton, 2009; Schofield et al., 2012a; Spacapan et al., 2017; Galland et al., 2019).

Thank you for this suggestion. We have expanded this paragraph to highlight the similarities between dykes and sills, and how evidence for sill structure and propagation processes can inform our knowledge of dykes. We have added references to several studies of sill geometry, from field observations and seismic reflection surveys.

Also some literature of the segmented nature of dykes and the associated complex magma flow could be added, either field-based studies (Poland et al., 2004; Schmiedel et al., 2021; Stephens et al., 2021) of analogue modelling studies (Takada, 1990; Kavanagh et al., 2018). This latter reference is particularly relevant, as Kavanagh et al. (2018) image a complex, localised analogue magma flow within a single dyke segment, i.e. similar to what the authors observe.

I thus invite the authors to better place the results of their study in the scientific context.

We have added more references to provide context for our study. Firstly, we have added mention of complex magma flow within segments (line 32) drawing attention to the variability in flow directions recorded at the margins of dyke segments, citing the work of Baer and Reches (1985), Poland et al. (2004, 2008), and Schmiedel et al. (2021). Secondly, we have added a short discussion around flow patterns within analogue intrusions of isothermal and solidifying liquids in the second paragraph of the introduction.

Finally, the third chapter of the introduction is rather speculative and vague. The authors introduce numerous potential implications of revealing the 3D structures of dykes and associated complex magma flow, but this paragraph contains only one citation, the rest being a list of statements not supported by the literature.

The relative paucity of citations in this section highlights that there is a gap in our conceptual understanding of dykes, and the impacts of their segmentation. It is evident

that fissure eruptions often involve multiple en-echelon fissure segments, and that these may display different behaviours, but to our knowledge there has been little to no work on how this surface expression of segmentation relates to the underlying dyke structure, or how the connectivity of segments may influence magma flow or the transport of volatiles. One of the key aims of our manuscript is to draw attention to this gap. Currently, attempts to model the processes beneath erupting fissures treat feeder dykes as simple, planar fractures experiencing vertically upwards flow, which is at odds with field and seismic evidence of complex, segmented structures. We want to highlight the fact that more work is needed in the area, firstly to establish the morphology of segments and complexity of conduits, and secondly, to explore the effects of segment morphology on eruptive processes.

Nevertheless, we appreciate the reviewer's concerns, and we have cited two additional studies in this section (starting line 47): firstly, an analogue study from Jones and Llewellyn (2019) exploring the organisation and localisation of convection within feeder dykes, which is partly controlled by segment length; and secondly, analogue studies from Menand and Phillips (2007) and Pioli et al. (2009) on how branching conduits can lead to gas segregation.

In general, the authors are honest by writing that they present a conceptual model. Nevertheless, this conceptual model includes magma flow dynamics, rock mechanics for dyke propagation, and thermal feedback, i.e. quite a complex set of processes that interplay. However, the data provided by the authors only support the magma flow dynamics, but the authors provide little evidence of thermal effects, and no structural data to discuss rock mechanics associated with dyke propagation. As such, this conceptual model is very speculative.

We appreciate that our model is conceptual, and that it involves a complex set of interrelated processes. However, a key aim of our manuscript is to highlight that the current approach to modelling dykes involves handling these processes in isolation. For example, dyke propagation models focus on rock mechanics and neglect cooling and solidification, even though the narrow leading edge must be vulnerable to these processes. By contrast, thermal models neglect rock mechanics and assume a pre-existing conduit where magma is already flowing. While these models are undeniably valuable, neither can capture dyke tip processes. This is a gap in understanding that needs to be addressed, because dyke tip processes determine the direction of propagation, the development of segments, and the likelihood of stalling versus eruption. A model capturing the dynamics of these processes is beyond the scope of this work, so our aim is to prompt

a discussion around segmentation processes, the types of evidence we can use to infer them, and their importance in governing emplacement and eruption dynamics.

We respectfully disagree that our field data only supports magma flow dynamics. The margins of dykes are evidence of dyke tip processes, because the margins are comprised of magma from the dyke tip. The shape of the margins shows how the magma interacted with the host rock when the dyke tip arrived; indeed, the margins can provide a better insight into tip processes than solidified tips themselves, which can be modified by inflation/deflation (e.g., Daniels et al., 2012; Schmiedel et al., 2021). On the Carrizales Dyke, the margins are undulous, with the magma appearing to have pushed into gaps between the more resistant clasts of the host rock. This captures the earliest mechanical interactions between the dyke and its host, and demonstrates that it wasn't a clean, straight-edged fracture. Instead, it was convoluted, with highly localised variations in width that would have impacted subsequent flow and cooling dynamics.

The reviewer's comment makes it apparent that we were insufficiently clear in our message regarding margin shape, host rock interaction and dyke tip processes. **As such, we have added additional photographs of the Carrizales Dyke margin (Fig. 3), and expanded our explanation and interpretation of margin shape (line 236) to emphasize the message that we summarize above.**

The reviewer also mentions a lack of evidence for thermal effects, and a lack of structural data. We will address this in our responses to the comments below.

Section "Intrusion events"

Lines 141-142. The authors write "There is a chilled margin between the intermediate and central layers, with a noticeable drop in grain size." This statement is not supported by data. Please provide data that supports this statement.

We have added photomicrographs to the supplementary information showing the internal chilled margin, which is visible by eye (Supplementary Figure 10a). We have also added plots showing the distribution of plagioclase microlite lengths in the 2 mm either side of the chilled margin, which show a marked reduction across the boundary (Supplementary Figure 10b).

We also note that the internal chilled margin is particularly visible due to a series of dendritic oxide crystals which grow inwards from it. These are similar to "comb crystals" described elsewhere, which are known to only grow against existing solid surfaces such as dyke walls or around xenoliths (e.g., Donaldson, 1977; Katz and Keller, 1981; McCarthy and

Müntener, 2016). A discussion on the growth of these dendritic crystals is beyond the scope of this study, but their presence makes us very confident that the intermediate layer of the Carrizales Dyke was already solid when the central layer was emplaced. However, the nature of the chilled margin has no bearing on segmentation or localisation dynamics, which was why we kept our description brief. We hope that the new supplementary figures address any concerns regarding the chilled margin.

Discussion

Section "Segmentation and stochastic propagation"

This section, and in particular the second paragraph, is quite speculative. Except the irregular shape of the dyke segments, the authors do not provide structural elements (fractures, faults, etc) that would feed this discussion. The second paragraph introduces and links several concepts and mechanisms, with only one citation to the literature (a paper of the authors themselves) and no reference to any data and figure in this manuscript. I find it too conceptual and speculative, and not enough supported by data and/or literature.

Please note that this section has now been incorporated into the “Inferred architecture and emplacement” sub-section (line 211).

The reason that we do not provide structural elements is because we saw no evidence for faults or fractures controlling the segmentation of the Carrizales Dyke. We have reworded this section to state this more clearly (line 231). Dyke D2 is the only large-scale structural control evident on the studied section of the Carrizales Dyke. We discuss how the inconsistent offset direction between segments makes it unlikely that they were guided by a rotation of the regional stress field in the direction of propagation, as predicted by Pollard et al. (1975). So, segmentation was guided by some other process.

We view the lack of large-scale structural controls as an important observation, providing insight into segmentation processes. For example, there are no fractures within the host rock separating S2 and S3 (Fig. 1e), and no fractures are present in the host rock where S3 and S4 are in contact (Fig. 1d). Therefore, if fractures did not guide segmentation, what did? We are forced to be speculative, because the usual culprits for segmentation are not present. This is why we discuss the margins, which are locally irregular, deflecting around clasts in the host rock, because they provide evidence for dyke-tip processes. As we discussed earlier, it is the dyke tip that “chooses” the propagation path, opening a new fracture in the most energetically favourable direction. So, in the absence of large

fractures, faults, or lithological boundaries, we question whether segmentation can be triggered by small-scale features, easily overwritten by later inflation.

To make our message clearer, we have reworded the second paragraph to explore the relation between margins and dyke tip processes in more detail (line 236). The undulous margins show that the initial fracture was convoluted, rather than having straight edges, which would have led to variations in magma flow rate. We have added references to studies of dyke cooling, showing that a narrow leading edge would cool quickly, making it highly susceptible to thermal instabilities. The dyke tip is therefore very likely to experience spatial and temporal variations in magma flow rate – and there is physical evidence for this in the banded margins. Our previous work, which we cited here, included numerical modelling of cooling timescales, pressure fluctuations, vesicle growth, and flow-induced phenocryst organisation, to show that banding at dyke margins, as seen on the Carrizales Dyke, can be explained by pulsatory flow at the dyke tip. Taken together, the evidence from the margins of the Carrizales Dyke points towards a staggered, localised advance of the leading edge.

Section "Thermal feedback"

This section discusses the potential thermal effects of magma flow localisation. Again, I find this discussion too qualitative and not enough supported by the literature or data. One can also note that the authors have not conducted any thermal calculations, even in orders of magnitude, to support their discussion.

The main argument of the authors supporting thermal effects is the observation of magma flow localisation. However, Kavanagh et al. (2018) show that localised magma flow can occur without thermal effects. Thus, the authors need more observations/data/analyses to feed this discussion.

Please note that we have moved the main points from the “Thermal feedback” sub-section into the new “Sub-layering and flow localisation” sub-section (starting line 160). We also discuss the findings of Kavanagh et al. (2018) in this sub-section.

This section is qualitative due to the enormous complexity involved in thermal calculations. There is currently no model that can fully describe the localisation dynamics of a cooling, solidifying flow, even within a rectangular channel, and developing one is far beyond the scope of this study. Again, this is another significant gap in our understanding of dyke dynamics, and we want to draw attention to this – especially in the context of the dyke tip, which has been neglected in all thermal modelling studies to date. We have

added references to numerical models for rectangular channels and 2D systems, which are typically used in the context of feeder dykes directly below fissure eruptions, and we suggest that the same processes must be in operation during the propagation phase, enhancing existing localisation mechanisms (line 174).

Section "Conceptual model for dyke emplacement"

This section is probably the most problematic. It develops a conceptual model of the potential processes that may be at work at the leading edge of the studied dyke. The problem here is that there is no observation of the leading edge of the dyke, therefore this whole discussion is poorly linked to the study (very few references to the data), and can almost be written as a general, simplified review of potential processes at the leading edge of a dyke. Thus, the data provided in this study barely support this discussion, which appears somehow off-subject. In addition, the third and fourth paragraphs of this section contain almost no reference to the literature (only one) and references to qualitative observations. These paragraphs appear as a list of statements, which provide the feeling to the reader that they are quite speculative. This is not sufficient for publication in Nature Communications.

The greatest obstacle to all studies of dyke propagation is that we have never observed the active leading edge, or "dyke tip". However, processes at the leading edge are the first control over dyke emplacement, and they determine the final, solidified product. The shape of the margins, the organisation of segments, and the localisation of flow are *all* influenced by dyke tip processes. They are all products of the leading edge, from the initial conditions within the earliest fracture through the host rock. As such, we do not see this discussion as being "off-subject". The crucial, early moments of dyke emplacement are consistently neglected or simplified by numerical models, and it is vital that we discuss the implications of this. No numerical model has studied the dynamics of segmentation, yet field observations and seismic evidence show that segments are an intrinsic part of dyke emplacement. Our discussion is intended to highlight knowledge gaps, and to raise questions. All the processes discussed here have a physical basis, and have been applied to other parts of the dyke system (e.g., the established conduits beneath fissure eruptions), but they have never been considered in the context of the dyke's leading edge. We want to demonstrate that the causes and implications of segmentation are a vital, under-studied aspect of dykes and associated fissure eruptions.

To strengthen the physical and evidential basis for our arguments, we have included more citations in this section, referencing works that were discussed earlier in the manuscript,

including observations from sills, thermal models, etc. We have also reworded some sections to emphasize links with field evidence presented earlier in the manuscript, and links to other dyke studies.

All in all, I feel that this study is very interesting, the figures are very elegant, the data presented are very sound, but the ambitious conclusions and conceptual model are not supported by the data. This makes the study, as it is, not suitable for publication in Nature Communications. I thus recommend the authors to considerably strengthen their data set, or to reduce the ambition of the conclusions of the study.

We agree that the conclusions are ambitious, but we respectfully disagree that they are not supported by the data. We hope that the expanded dataset in the supplementary file, and the addition of more references to sills, analogue experiments and thermal models, have strengthened the data and literature basis for our interpretations, and better demonstrated the context of our study. We also hope that our ambitious conclusions and conceptual model will spur further research in this area, to bring further quantitative rigour to bear on testing our hypotheses.

Response to reviewer 2:

In their manuscript “Localisation during dyke propagation produces complex magma transport pathways”, Allgood et al. present data on the external and internal architecture of a basaltic dyke exposed in the Teno massif of Tenerife, Canary Islands, Spain. They used field observations, image analysis and thin section analysis to derive information on magma flow during dyke emplacement. They then use this data to infer a model of dyke propagation that resembles the development of lava flow fronts, including lobes and channelized flow. The study presented in the manuscript relies on quantitative data and comes up with a novel view on dyke propagation, which should be of interest to the readers of Nature Communications.

The manuscript is well written and clear. However, there are a few major points that need to be considered in a revised version of the manuscript. All my comments are listed below, but the major ones to focus on are clearly 2, 5/6, 7, 12, 14, 15 and 16. More specifically, (a) I miss quantitative data on dyke segment geometry and pre-existing structures. (b) I miss information on the host rock and minerals. (c) I want to see discussion of alternative explanations regarding flow localisation (other than thermal effects) and downward magma flow. Moreover, I suggest adding a wider selection of references and gave some examples below.

Kind regards,

Steffi Burchardt

List of suggested improvements

1. Title: I suggest adding “flow” to the title to read “flow localization during dyke propagation” as just “localisation” is too unspecific.

A similar point was made by Reviewer 1. We agree and we have changed the title to specify magma flow localisation, to avoid any confusion. Thank you for the suggestion.

2. Line 45: the Tajogaite eruption on La Palma occurred in 2021 not 2022.

Thank you for spotting this typo! It has been fixed.

3. More information on the host rock is needed. The authors mention that the host rock is volcanoclastic and infer that it was potentially unconsolidated which affected dyke propagation. However, the rock type is not described in enough detail. Also, from the field photographs it appears that the host rock varies from massive (e.g. Suppl. Fig. 2a) to layered (e.g. Suppl. Fig. 2b). Some researchers studying magmatic sheet propagation are of the strong opinion that the host rock has a considerable effect on sheets (e.g. Schmiedel et al., 2017; Vachon & Hieronymus, 2017; Galland et al., 2019), whereas others think it the host rock is secondary to other parameters (e.g. Schmiedel et al., 2021; Stephens et al., 2021). Also, there is literature on dyke propagation through unconsolidated material that should be discussed and cited (Petronis et al., 2013).

The host rock is volcanoclastic all along the ridge. We have added a new paragraph at the end of the sub-section “The Carrizales Dyke” that outlines the sizes and compositions of clasts, which are predominantly lava or pyroclastic material (line 91). We do not think that a discussion of propagation through unconsolidated material would add anything to our main interpretations, because we cannot know the state of the host rock when the dykes intruded.

In Supplementary Figure 2a, the rock at the dyke margin is not the original host rock, because a second, younger dyke has intruded alongside the Carrizales Dyke for a short stretch (around 15 m). This younger dyke cuts across the Carrizales Dyke several metres from where this photo was taken, and as it is evidently younger, and could not have influenced the emplacement of the Carrizales Dyke, we did not discuss it further in the main text.

4. Fig. 1: Photographs are too small/dark. While I appreciate that the photos are available in a larger format in the supplementary, I think most readers will not go there. The photographs in Fig. 1 (c-e) are too small, the lines are hard to see (especially the green and blue ones), and e is far too dark. I don't think Nature Communication restricts the size of figures so much. Also, I would prefer to have an inset with the location data in Fig. 1 instead of the supplementary.

Figures have been resized to meet journal requirements. Figure 1 is now a two-column figure, and the size of the photographs has been increased. However, we do not want to overcomplicate Figure 1 by adding inset figures. The precise location of the dyke does not influence the conclusions of our study, so we did not think it was necessary to include a map of Tenerife in the main figures.

5. Dyke morphology: the morphology of magmatic sheet intrusions has been used extensively to derive information on sheet propagation. Since, the Carrizales dyke is so well exposed, it would be useful to present this data (segment length vs. thickness vs. tip geometry) and put it into a scientific context (see Schmiedel et al., 2021). The authors mention in the Methods section that segment lengths and thicknesses were measured, but apart from Fig. 1 and an average thickness, this data is neither presented nor discussed. The thickness along the segment would be particularly interesting for segment 4, where it should be related to the emplacement of the lenticular layers 3b and 3c.

Segment lengths are now shown in Figure 1, and provided as a supplementary data file (an Excel spreadsheet). The only segment accessible along its entire length is S4, and widths from along this segment were used to construct Figure 4. We realise that in the first version, we didn't explain that every vertex in Figure 4 represents measured layer and sub-layer widths, so this is now explained explicitly in the caption, and in the Methods section. All widths measured along the dyke are included in the supplementary Excel file.

We do not think that final dyke width has any implications for the emplacement of the nested sub-layers 3b and 3c. S4 has a relatively constant width for most of its length – it is not wider or narrower where the nested sub-layers are found. This is likely because solidification times depend on magma flow rate, rather than just dyke width (e.g., Lister and Dellar, 1996). Faster flowing sections, where localisation is likely to occur, will remain molten for longer, without being wider.

We have also added a new, short paragraph regarding the additional information that can be gleaned from tip geometries (line 126). Due to the dyke cooling, solidifying, inflating and deflating, and adopting a complex form due to heterogeneities in the host rock, the solidified state does not represent the molten state (Kavanagh and Sparks, 2011; Daniels et al., 2012). It is hard to relate solidified tips to active tip processes, and they must be interpreted as capturing a stage during emplacement, after the tip stalled and started to inflate (e.g., Stephens et al., 2021; Schmiedel et al., 2021). Given that some degree of inflation must have occurred, our interpretation is that the active tips at the dyke leading edge must have been even narrower and more tapered than the solid, inflated tips we see today. This supports our interpretations of the undulous dyke margins, which also provide evidence for dyke tip conditions.

6. Three-dimensional architecture: *The study of the 3D architecture of the Carrizales dyke would have been greatly aided by photogrammetry and 3D visualisation (e.g. Schmiedel et al., 2021). Also, quantitative data on segment geometries, overlap etc. such as in Delaney et al. (1986) should be presented and discussed. Overall, the section “Three-dimensional architecture” is not really what it promises. Instead, it describes the relationship between segments in a qualitative way. Additionally, since there is a later section called “inferred architecture” the naming of the sections becomes confusing. I suggest to rename the “Three-dimensional architecture” section to something like “segment geometry.”*

Quantitative data on segment overlaps are now presented in Figure 1, and in the supplementary Excel file. In the re-structuring of the manuscript to meet journal requirements, we have renamed the “Three-dimensional architecture” sub-section as “Segment geometry and inferred propagation directions”, which more accurately reflects the content. Throughout this section, we have added quantification of apparent lengths and widths (e.g., lines 109, 114).

7. Flow directions: *It would be great to add an example photograph of the imbricated phenocrysts and of the imbricated microlites in thin section to Fig. 4. Please also add the microstructural images to the Supplementary information. Also, what minerals were considered? What type of crystal habitus do they have? Crystal shapes (whether needly or platy) and phenocryst arrangements (single crystals or glomerocrysts) strongly affect the alignment. For instance, magma flow in the Sandfell laccolith in Iceland is outlined (in part) by the alignment of single, platy plagioclase phenocrysts (Mattsson et al., 2018). Likewise, needly hornblende phenocrysts align in the Cerro Bayo cryptodome, but plagioclase*

phenocrysts do not (Burchardt et al., 2019). The latter because plagioclase occurs as glomerocrysts.

We have added photomicrographs to the supplementary information (Supplementary Figures 6-9). An example photomicrograph showing plagioclase microlite imbrication is now also included in Figure 4.

In terms of minerals and fabrics, the dominant microlite phase is plagioclase, which has a platy form, as evidenced by the elongated cross-sections captured in orthogonal planes (Supplementary Figures 6-9). However, while the *amount* of crystal alignment may be influenced by crystal shape, the *direction* of alignment will not. So, for our purposes of showing an absolute flow direction, the shape of the crystals has only minor importance. Any elongated shape, whether platy or needle-like, would produce imbrication pointing the same direction, and all the crystals here, whether they are plagioclase microlites or clinopyroxene phenocrysts, indicate SW lateral flow.

8. *Figure 3: Please add a legend instead of describing the layers in the figure caption. Also consider add a close-up photo of each layer.*

An additional figure has been added (Figure 3) to demonstrate the nature of the three layers and sub-layers.

9. *Figure 4 and Suppl. Figs. 4a-d and Suppl. Fig. 5. Each rose diagram needs to be labelled with the amount of data displayed.*

Values have been added, showing the total amount of data in each rose diagram.

10. *Suppl. Fig. 3a. What is the yellow dashed line? (I guess it is the dyke margin as mentioned in the caption to Suppl. Fig. 3b. But please also add to 3a.*

Yellow dashed line is now labelled as dyke margin.

11. *Segmentation and stochastic propagation: “Stochastic” is not a term that has been used in the intrusion community so far. Please introduce it with a definition.*

The term “stochastic” is used widely in the natural sciences, especially in physics and mathematics. We are confident that most readers of Nature Communications will be

familiar with the term. The sub-section heading that included this term has been removed by the restructuring.

12. Thermal feedback: *Analogue models of dyke propagation by Janine Kavanagh and her group show flow localisation due to fluid dynamics within the intrusion and not related to thermal effects (e.g. Williams et al., 2022; Chalk and Kavanagh, 2024). While actual dykes of course must experience the effect of magma temperature, thermal feedback might not be the only mechanism responsible for flow localisation. I therefore suggest to discuss alternatives, such as fluid dynamics and even shear thinning rheology of magma.*

We have removed the section on thermal feedback in favour of incorporating it into “Sub-layering and flow localisation”, where we discuss other potential mechanisms, referencing the work from Kavanagh’s group.

13. Conceptual model, Line 239ff: *“We note that highly localised propagation is recorded at the margins of some dykes⁴, and that pulsatory propagation has been inferred from studies of other intrusions^{25,30}.” This sentence sounds to me like “other intrusions” should be intrusions that are not dykes. However, both citations refer to dyke studies. Pulsatory magma emplacement seems to be the norm these days and has been recognised in sills (e.g. Köpping et al., 2022), laccoliths (e.g. Witcher et al., 2024) and plutons (e.g. Menand et al., 2011). Also, I would suggest to link back to the seismic monitoring of dyke propagation here, with the Holuhraun dyke as a prime example of a dyke that propagated in segments (Sigmundsson et al., 2015).*

We have added references to pulsatory propagation in sills, which are the most comparable to dykes in this context, and to seismic data showing propagation in segments/pulses (line 259).

14. Downward magma flow: *The data presented in this manuscript are interpreted indicating downward flow of magma. However, fabrics in solidified dykes are not recording primary, i.e. emplacement-related, magma movement. Since the propagation stage is usually followed by hours to weeks of magma flow through a fissure before flow localises to vents in fissure eruptions, other processes can affect magma flow directions. For instance, dyke closure, sagging, and consolidation may alter the fabrics that will be preserved in the rock (see Andersson et al., 2016 Discussion and Fig. 8). Sagging may*

explain the downward flow indicators in the Carrizales dyke and should be discussed, since the conceptual model relies on the downward flow to be primary.

We are confident that the marginal textures record primary flow directions. Firstly, we note that imbrication in the horizontal plane is stronger than in the vertical plane, suggesting that the dominant flow direction was dominantly lateral, with a downwards component. During sagging or drain-back at the end of emplacement, flow will be dominantly downwards. This would overwrite any pre-existing lateral fabrics. Therefore, the fact that the lateral fabrics remain dominant in each layer suggests that sagging was minimal, and that the fabrics were captured while the magma was flowing, or shortly after it ceased.

Secondly, we see downwards imbrication at the margins of the dyke. The dyke margins originate at the leading edge, solidifying within minutes to hours of magma entering the crack, as shown by the thermal modelling in Allgood et al. (2024). As such, the margins capture the initial flow directions. We have added swaths of microscope images from the margin inwards to the supplementary information (Supplementary Figures 7c,7d), showing that downwards imbrication is present at the margin, and is dominant moving inwards (although we note that the vertical flow fabric is weaker in intensity than the horizontal fabric, suggesting that the dominant component was horizontal). So, we are confident that the initial magma pulse was lateral with a downwards component. It is likely that subsequent pulses, utilising the same conduit, and comprising the intermediate and central layers, flowed the same way. The fact that the three layers all contain textures imbricated in the same manner indicates that the flow direction remained constant, presumably fed from the same source region, until each layer solidified.

We have added a paragraph with a brief discussion of these points (line 204) and are grateful for the opportunity to clarify this issue.

15. Magma flow studies on dykes in similar settings: *The authors should discuss their findings in the context of other studies of magma flow in dykes, particularly in volcanic rift zones. Several papers focus on fabrics that indicate magma flow in Canary Island dykes, e.g. Delcamp et al. (2015). In this paper, the authors discuss the effect of topography causing downward flow along the rift zone.*

We have added a paragraph discussing our findings in the context of previous studies of laterally propagating dykes on Tenerife (line 212). However, we add the caveat that we cannot know if the overall propagation direction was downwards towards lower elevations, as inferred in previous studies (Soriano et al., 2008; Delcamp et al., 2015). The downwards

flow component may well be local, occurring at the lower edge of a segment that propagated dominantly laterally, as we show in Figure 5.

16. Pre-existing structures: *The Teno massif is the remnant of a volcanic rift zone. Dyke emplacement there was therefore strongly controlled by tectonic stresses, topography, and pre-existing structures, such as faults (cf. Delcamp et al., 2012 on the Anaga massif). Dykes tend to follow pre-existing weaknesses that have a favourable orientation in the prevailing stress field at the time of emplacement. Also, flow localisation is more likely where several fractures intersect. Therefore, it is important to provide data on the orientation of the dyke and other structures in the vicinity (e.g. Greiner et al., 2023). The only other structure that is mentioned is D2. But can the existence and influence of fractures really be ignored? The photos in Suppl. Figs. 2b and c show fractures sub-parallel to the dykes.*

We agree that the large-scale orientation of the Carrizales Dyke has developed in response to regional stresses. However, these regional-scale dynamics are not the focus of our study, so we do not discuss them in depth. Instead, we focus on dyke tip processes and the effects of segmentation.

The host rock contains fractures, but these are highly localised, and we cannot know their ages relative to the Carrizales Dyke. For example, there are two prominent fractures in Figure 1e, but these are restricted to a 15-m-thick layer of volcanoclastic material, show no significant vertical offset, and do not interact with any of the dykes within this cliff. Figure 1c shows a similarly localised fracture on another cliff, again with insignificant vertical offset, but this fracture also cuts across another dyke (to the right of “Other dykes” annotation). It is evident that we cannot know the relative ages of these fractures, and that they may have formed long after dyke emplacement. Furthermore, we saw no evidence of the Carrizales Dyke interacting with fractures or faults, a point which we have emphasized further in line 231 (also, see our response to the comments of Reviewer 1). As such, we do not believe that a detailed description of nearby fractures is warranted.

References in response letter:

Daniels, K.A., Kavanagh, J.L., Menand, T. and R. Stephen, J.S. (2012) The shapes of dikes: Evidence for the influence of cooling and inelastic deformation. *GSA Bulletin*, 124(7-8), 1102-1112. <https://doi.org/10.1130/B30537.1>

Delaney, P.T. and Pollard, D.D. (1981) Deformation of host rocks and flow of magma during growth of minette dikes and breccia-bearing intrusions near Ship Rock, New Mexico. *USGS Professional Paper*, 1202, 59-61. <https://doi.org/10.3133/pp1202>

Delcamp, A., Petronis, M. S., & Troll, V. R. (2015) Discerning magmatic flow patterns in shallow-level basaltic dykes from the NE rift zone of Tenerife, Spain, using the Anisotropy of Magnetic Susceptibility (AMS) technique. *Geological Society, London, Special Publications*, 396, 87-106. <https://doi.org/10.1144/SP396.2>

Donaldson, C.H. (1977) Laboratory duplication of comb layering in the Rhum pluton. *Mineralogical Magazine*, 41(319), 323-336. <https://doi.org/10.1180/minmag.1977.041.319.03>

Katz, K. and Keller, J. (1981) Comb-layering in carbonatite dykes. *Nature*, 294(5839), 350-352. <https://doi.org/10.1038/294350a0>

Kavanagh, J.L. and Sparks, R.S.J. (2011) Insights of dyke emplacement mechanics from detailed 3D dyke thickness datasets. *Journal of the Geological Society*, 168, 965-978. <https://doi.org/10.1144/0016-76492010-137>

Lister, J.R. & Dellar, P.J. (1996) Solidification of pressure-driven flow in a finite rigid channel with application to volcanic eruptions. *Journal of Fluid Mechanics*, 323, 267-283. <https://doi.org/10.1017/S0022112096000912>

McCarthy, A. and Müntener, O. (2016) Comb layering monitors decompressing and fractionating hydrous mafic magmas in subvolcanic plumbing systems (Fisher Lake, Sierra Nevada, USA). *Journal of Geophysical Research: Solid Earth*, 121(12), 8595-8621. <https://doi.org/10.1002/2016JB013489>

Schmiedel, T., Burchardt, S., Mattsson, T., Guldstrand, F., Galland, O., Palma, J.O. and Skogby, H. (2021) Emplacement and segment geometry of large, high-viscosity magmatic sheets. *Minerals*, 11(10), 1113. <https://doi.org/10.3390/min11101113>

Soriano, C., Beamud, E. & Garcés, M. (2008) Magma flow in dikes from rift zones of the basaltic shield of Tenerife, Canary Islands: implications for the emplacement of buoyant magma. *Journal of Volcanology and Geothermal Research*, 173(1-2), pp.55-68. <https://doi.org/10.1016/j.jvolgeores.2008.01.007>

Stephens, T., Walker, R., Healy, D. and Bubeck, A. (2021) Segment tip geometry of sheet intrusions, II: Field observations of tip geometries and a model for evolving emplacement mechanisms, *Volcanica*, 4(2), 203–225; <https://doi.org/10.30909/vol.04.02.203225>

We would like to thank both reviewers for providing insightful comments that have greatly improved the manuscript. There was one outstanding comment from Reviewer 2 in their final review, and we have now addressed this.

“More information on the host rock and minerals. Mostly done. However, I suggested to cite and discuss literature on dyke propagation through unconsolidated material (Petronis et al., 2013), which the authors did not take on board, because one cannot know the state of the host rock at the time of dyke propagation. Yet, they state that the dyke “intruded when the volcanoclastic host rock was relatively unconsolidated.” Also, a lot of the interpretation relies on the interaction of the dyke tip with the irregularities in the host rock. So I encourage the authors to address this comment.”

To address this comment, we have added a brief discussion regarding the state of the host rock and how the dyke tip might have interacted with it (Section “The Carrizales Dyke”, second paragraph). We note that the volcanoclastic host rock may have been relatively unconsolidated, as suggested by the undulous margins, but it was evidently still consolidated enough to support broadly planar cracks, as we see no evidence of the extreme dyke width variations associated with intrusions into fresh, unconsolidated scoria cones, as reported by Petronis et al. (2013). Due to the host rock being coarsely clastic and unconsolidated to an unknown degree, we have added clarification that the dyke may have intruded through a combination of brittle fracture and viscous indentation, similar to the mechanisms suggested by Stephens et al. (2021). However, whether the dyke intruded by viscous indentation or brittle fracture, the subsequent cooling and thermal feedback processes driving localisation within segments would operate in the same manner.

Stephens, T., Walker, R., Healy, D. & Bubeck, A., Segment tip geometry of sheet intrusions, II: Field observations of tip geometries and a model for evolving emplacement mechanisms, *Volcanica*, 4(2), 203–225, (2021).

<https://doi.org/10.30909/vol.04.02.203225>